# Safety and Efficacy of Felid Herpesvirus-1 Deletion Mutants in Cats

**DOI:** 10.3390/v13020163

**Published:** 2021-01-22

**Authors:** Yao Lee, Roger K. Maes, John M. Kruger, Matti Kiupel, Kim S. Giessler, Gisela Soboll Hussey

**Affiliations:** 1Department of Pathobiology and Diagnostic Investigation, College of Veterinary Medicine, Michigan State University, 784 Wilson Road, East Lansing, MI 48824, USA; leeyao@msu.edu (Y.L.); maes@dcpah.msu.edu (R.K.M.); kiupel@msu.edu (M.K.); giessle1@msu.edu (K.S.G.); 2Veterinary Diagnostic Laboratory, Michigan State University, 4125 Beaumont Road, Lansing, MI 48910, USA; 3Department of Small Animal Clinical Sciences, College of Veterinary Medicine, Michigan State University, 784 Wilson Road, East Lansing, MI 48824, USA; krugerj@msu.edu

**Keywords:** Felid herpesvirus-1, glycoprotein E (gE), thymidine kinase (TK), serine/threonine protein kinase (PK), vaccination, cats

## Abstract

Felid herpesvirus-1 (FeHV-1) is an important respiratory and ocular pathogen of cats and current vaccines are limited in duration and efficacy because they do not prevent infection, viral nasal shedding and latency. To address these shortcomings, we have constructed FeHV-1 gE-TK- and FeHV-1 PK- deletion mutants (gE-TK- and PK-) using bacterial artificial chromosome (BAC) mutagenesis and shown safety and immunogenicity in vitro. Here, we compare the safety and efficacy of a prime boost FeHV-1 gE-TK- and FeHV-1 PK- vaccination regimen with commercial vaccination in cats. Cats in the vaccination groups were vaccinated at 3-week intervals and all cats were challenge infected 3 weeks after the last vaccination. Evaluations included clinical signs, nasal shedding, virus neutralizing antibodies (VN), cytokine mRNA gene expression, post-mortem histology and detection of latency establishment. Vaccination with gE-TK- and PK- mutants was safe and resulted in significantly reduced clinical disease scores, pathological changes, viral nasal shedding, and viral DNA in the trigeminal ganglia (the site of latency) following infection. Both mutants induced VN antibodies and interferons after immunization. In addition, after challenge infection, we observed a reduction of IL-1β expression, and modulation of TNFα, TGFβ and IL10 expression. In conclusion, this study shows the merits of using FeHV-1 deletion mutants for prevention of FeHV-1 infection in cats.

## 1. Introduction

Felid herpesvirus-1 (FeHV-1), a member of the alphaherpesvirinae, is an important viral pathogen of cats worldwide. The clinical signs associated with FeHV-1 infection include fever, rhinotracheitis, pneumonia, conjunctivitis, keratitis, neonatal fatality, and potentially facial and nasal dermatitis and abortion [1,2]. It is estimated that FeHV-1 accounts for approximately 50–75% of all diagnosed viral upper respiratory infections in cats [3] and because of this prevalence, FeHV-1 vaccines are part of the feline core vaccines [4]. Current recommendations by the World Small Animal Veterinary Association (WSAVA) are to vaccinate kittens against infection with FeHV-1 at 6–8 weeks of age, followed by at least one additional administration 2–4 weeks later [4]. Annual revaccination should be carried out in high-risk cats, such as pregnant queens or cats living in multi-cat environments.

Both inactivated and modified live virus (MLV) feline herpesvirus vaccines are widely used in the United States, but protection from FeHV-1 infection is limited in efficacy and shorter in duration compared to feline calicivirus (FCV) and feline panleukopenia virus (FPV), the other two components of feline core vaccination. Currently used FeHV-1 vaccines reduce the severity of clinical signs, but they do not prevent FeHV-1 infection itself in previously vaccinated cats and, as a consequence, cannot curtail establishment of FeHV-1 latency [3,5]. Environmental, physiological and chemical stressors can all lead to reactivation from latency and, as such, are associated with renewed replication and shedding of infectious virus [6,7]. This implies that clinically normal cats can shed virus and cause disease in unvaccinated animals, explaining the high incidence in shelters. Reactivation also plays an important role in immune-mediated dendritic keratitis [3].

In addition to the inability to prevent infection, mucosal administration of current vaccines labeled for subcutaneous (SC) use is not safe because the basis of the attenuation is the natural temperature sensitivity of FeHV-1. Thus, current MLV vaccines contain the complete set of viral genes that are involved in replication, virulence and immunomodulation but are attenuated at body temperature. However, at the reduced temperature of the upper respiratory tract, the vaccine virus retains its virulence. Reports on virulence of the vaccine virus in cats intranasally (IN) immunized with a commercial vaccine labeled for SC use, or in cats with inadvertent mucosal exposure following subcutaneous administration, are, therefore, not unexpected [4,8,9].

Despite widespread vaccination, FeHV-1 continues to circulate. This emphasizes the need for a vaccine that provides more comprehensive protection than achieved with the current vaccination regimen. Further, it is likely that stimulation of mucosal respiratory immunity plays an important role in stimulating responses that are crucial for more comprehensive protection. Molecular biology and recombinant DNA technology have made it possible to advance vaccine technology; for example, by generating safe gene-deleted viruses that can be administered either parenterally or mucosally or as a combination of both routes. A further advantage of gene-deleted vaccines is that they are “marker” vaccines, allowing for a distinction between vaccine virus and circulating field virus and the immune responses elicited by vaccination versus field virus infection. This is crucial aspect of eradication programs, such as the one that was successfully implemented for the control of pseudorabies virus (PRV) [10]. The generation of gene-deleted viruses has previously led to mutants that could be used as vaccines for pseudorabies virus, bovine herpesvirus-1 (BHV-1), and equine herpesvirus-1 (EHV-1) [11,12,13]. In particular, the development and very successful use of a PRV gE-deletion mutant to protect against Aujeszky’s disease in pigs was a motivator for the development of next-generation vaccines for animal use [10]. Furthermore, Kimman et al. reported a combined in vitro and in vivo PRV study, which encompassed multiple deletion mutants, including a gE-deletion mutant (gE-), an Us3-encoded serine/threonine protein kinase-deletion mutant (PK-), a thymidine kinase-deletion mutant (TK-), a gE-PK-double-deletion mutant (gE-PK-), and a gE-TK-double-deletion mutant (gE-TK-). Pigs inoculated intranasally with the gE-PK- or gE-TK- mutants excreted low amounts of virus, but the levels of virus shedding in those inoculated with the gE-, PK-, and TK- single-deletion- mutants and wild-type (WT) raised concerns about the actual safety of the single-deletion mutants. In terms of protection against challenge virus replication, pigs immunized with the PK-mutant showed complete protection [11]. In cattle, experimental immunization with mutants with deletions of gE (gE-), TK (TK-), or both gE and TK (gE-TK-) induced immunity, which protected calves from BHV-1 infection. Experimentally intranasal or intramuscular vaccination with gE-deletion-mutants drastically reduced viral shedding and alleviated the symptoms of clinical disease after WT EHV-1 challenge in foals [13].

In cats, gI-gE- or TK-deletion mutants of FeHV-1 have been generated previously and shown promising results [14,15]. All of the above studies suggest that virulence factor-associated gene-deletion mutants not only have the potential for reduced virulence compared to their parent strain, but also to retain the level of antigenicity required to protect against both clinical signs and infection itself.

Based on this data, we have previously described the generation of gE-, PK-, gC- and gE- TK- deletion mutants of FeHV-1 by using bacterial artificial chromosome mutagenesis (BAC) [16,17]. We then extensively characterized these mutants in Crandell-Rees feline kidney (CRFK) cells, feline respiratory epithelial cells (FRECs) and tracheal explants [18,19]. Results in these systems showed that the PK-single-deletion mutant and the gE-TK-double deletion mutant exhibited reduced virulence, tissue damage and spread to the underlying stroma, while maintaining the ability to induce respiratory epithelial immunity. Consequently, we selected these two mutants for further evaluation. The purpose of the current study was to test whether we could use the selected FeHV-1 deletion mutants to develop a safe and efficacious subcutaneous prime-mucosal boost vaccination regimen that would be superior to currently used commercially FeHV-1 vaccination regimens in terms of reducing clinical signs, limiting virus shedding and latency, and inducing immunity in specific-pathogen-free cats subsequently challenged with virulent FeHV-1.

## 2. Materials and Methods

### 2.1. Animals

A total of 20 male domestic short hair specific-pathogen-free (SPF) cats were purchased from Marshall BioResources (https://www.marshallbio.com). All cats were 3.5 months old on arrival. Each group was housed in a separate room and cats within each group were housed in individual cages to avoid cross-contamination. All cats were acclimatized for 7 days and fed a diet of dry and moist food and had access to ad libitum water throughout the study. All cats received environmental enrichment in the form of human interaction not associated with sampling or diagnostics 2–3 time daily throughout the study. For procedures other than nasal swabbing (blood collection, vaccination, challenge infection), all cats were sedated with intramuscular injection of a combination of dexmedetomidine (30 mcg/kg), ketamine (5 mg/kg), and butorphanol (0.2 mg/kg). After procedures, sedation was reversed with intramuscular administration of atipamezole (0.2 mg/kg). All protocols performed in this study were approved by the Animal Care and Use Committee at Michigan State University (MSU), East Lansing, MI, USA.

### 2.2. Experimental Design

An experimental timeline of the study is shown in Table 1. Cats were randomly divided into 4 groups upon arrival: (1) Unvaccinated controls, *n* = 4; (2) Commercial vaccine group, *n* = 4, immunized subcutaneously (SC) on days 21 and 42, with a commercial adjuvant-free modified-live vaccine (PUREVAX Feline 3, Merial Inc., Athens, GA, USA) according to the manufacturer’s recommendations; (3) gE-TK- vaccinated group, *n* = 6, vaccinated SC with the gE-TK- mutant on days 0 (V1) and 21 (V2), followed by an intranasal (IN) boost on day 42 (V3); (4) PK- vaccinated group, *n* = 6, vaccinated SC with the PK- mutant on days 0 and 21, followed by an IN boost on day 42. All groups were challenge infected (CH) intranasally with 4 × 10^5^ TCID_50_ of virulent WT FeHV-1 on day 63. Clinical signs and body temperatures were monitored daily throughout the study. Nasal secretions were collected using Dacron swabs prior to the start of the study and on days 2, 21, 23, 42, 44, 46, 48, 50, 52, 62, 65, 67, 69, 71, 73, 75, 77, 81 of the study as well as prior to necropsy, and were placed in 1 mL of transport medium and stored at −80 °C until further processing. Venous blood for measurement of virus neutralizing antibody titers was collected into serum separator tubes on days 0, 14, 21, 35, 42, 56, 63, and 77. Tubes were centrifuged, and the serum was harvested and stored a −80 °C until further testing. In addition, 2.5 mL of whole blood for isolation of mRNA was collected in PAXgene^®^ blood RNA tubes (IVD) (PreAnalytix GmbH, Hombrechtikon, Switzerland) on days 0, 23, 44, and 65. Necropsies with gross examinations and tissue collections for histology, as well as evaluation of viral DNA presence in trigeminal ganglia were performed between days 83 and 86. On the day of necropsy, cats were initially sedated and then euthanized by intravenous injection of 85.9 mg/kg pentobarbital sodium.

### 2.3. Viruses

Deletion mutants of FeHV-1 included a double-deletion-mutant (gE-TK-) as well as a single-deletion-mutant (PK-) and were constructed using the full-length bacterial artificial chromosome (BAC) clone of the C27 strain of FeHV-1 [ATCC, VR-636, Manassas, VA, USA] and two-step Red-mediated recombination [17]. A low passage virulent clinical FeHV-1 isolate from the Veterinary Diagnostic Laboratory (VDL) at MSU [FeHV-1 1627] was used as the challenge virus on day 63 at a dose of 4 × 10^5^ TCID_50_ per cat via intranasal instillation in anesthetized cats. For virus propagation and titration, the Crandell-Rees feline kidney (CRFK) cell line (ATCC, CCL-94) was used. For SC administrations of the gE-TK- or PK- mutants a dose of 1 × 10^6^ TCID_50_ was used per cat. For IN administration on day 42 a dose of 1 × 10^6^ TCID_50_ per cat was used for the gE-TK- mutant and a dose of 5 × 10^5^ TCID_50_ per cat was used for the PK- mutant. The total dose was divided into 2 aliquots, 100 µL each, and administered dropwise into each nostril of cats that had been anesthetized. A lower dose was chosen for the PK- mutants because of the lower maximum titer of the PK- mutant in CRFK cells.

### 2.4. Clinical Signs and Body Weights

The scoring method used to quantitate clinical signs of FeHV-1 associated disease has been described previously [20] and is shown in Table 2. Briefly, clinical signs included conjunctivitis, blepharospasm, ocular discharge, nasal discharge, sneezing, nasal congestion, coughing, and fever. For clinical assessment, groups of cats were evaluated by 2 observers for 60 min daily. Clinical signs were quantitated, and scores were assigned to each cat for each day. Daily scores for each cat were then added for the time period after vaccination and after challenge infection and median scores were calculated for each group. This resulted in a median total clinical score for each group pre-vaccination, post-vaccination 1 (post-V1, day 0–21), post-vaccination 2 (post-V2, day 22–42), post-vaccination 3 (post-V3, day 43–63) and post-challenge infection (post-CH, day 64–83).

### 2.5. Real-Time PCR on Nasal Swab Extracts

Nasal swabs were collected into 1 mL of viral transport media (EMEM) (Sigma-Aldrich, St. Louis, MO, USA) supplemented with 1% penicillin-streptomycin (Life Technologies, Carlsbad, CA, USA). Total DNA was extracted from 200 μL aliquots of nasal swab samples using the DNeasy Blood and Tissue kit (QIAGEN, Hilden, Germany). Extracted DNA was quantitated with a Nanodrop spectrophotometer (Thermo Fisher Scientific, Waltham, MA, USA) and real-time PCR on 20 ng of extracted DNA was performed using a 7500 Fast Real-Time PCR System with 7500 Software v2.0.6 (Applied Biosystems by Life Technologies Corp., Austin, TX, USA) as previously described [21]. Briefly, primers and probe used were: Forward (5′-3′): AGA GGC TAA CGG ACC ATC GA; Reverse (5′-3′): GCC CGT GGT GGC TCT AAA C; Probe (5′-3′): FAM-TAT ATG TGT CCA CCA CCT TCA GGA TCT ACT GTC GT-BHQ-1. The PCR amplification parameters were 2 min at 50 °C and 30 s at 95 °C, followed by 40 cycles of denaturation at 95 °C for 15 s and annealing-elongation at 60 °C for 1 min. Each sample was analyzed in triplicate and a positive and negative control was included on each plate.

### 2.6. Conventional PCR to Differentiate between WT Virus and Mutant Viruses

In order to differentiate WT from the mutant viruses in nasal swabs, conventional PCR assays targeting FeHV-1 gE, PK, and TK genes were performed, as previously described [17]. Sequences of forward and reverse primers are listed in Table 3. A 20 μL PCR cocktail was composed of 10 μL GoTaq^®^ G2 Green Master Mix (Promega, Madison, WI, USA), 1.2 μL of forward primer (240 nM), 1.2 μL of reverse primer (240 nM), water 3.6 μL, and template 4 μL. PCR cycling conditions were 94 °C for 2min, followed by 40 cycles of 94 °C for 1 min, 55 °C for 1 min, and 68 °C for 1min, followed by final elongation at 68 °C for 10 min. Amplification products were resolved in 1% agarose gels prepared using 1X TBE (Bio-Rad, Hercules, CA, USA) and visualized using SYBR^TM^ Safe (Life Technologies, Carlsbad, CA, USA).

### 2.7. Virus Isolation

Nasal swab eluates were filter-sterilized using Millex syringe filters with an average pore diameter of 0.45 μm. (Millipore, Bedford, MA, USA). For a standard virus isolation assay, a total number of 2.5 × 10^4^ CRFK cells and 50 μL of filtrate were added to duplicate wells of a 96 well microtiter plate. The plates were incubated for 3 days at 37 °C, 5% CO_2_ and the presence or absence of characteristic cytopathic effect was evaluated for determining whether samples were positive or negative [22,23].

### 2.8. Evaluation of Viral Invasion in Trigeminal Ganglia

Trigeminal ganglia preserved in OCT compounds frozen at −80 °C were used for DNA extraction [24]. Approximately 15 mg of tissues were lysed in 180 μL buffer ATL and 20 μL proteinase K at 65 °C overnight (QIAGEN, Hilden, Germany), followed by extraction using the DNeasy Blood and Tissue kit (QIAGEN, Hilden, Germany) according to manufacturer’s instructions. Extracted DNA was quantitated with a Nanodrop spectrophotometer (Thermo Fisher Scientific, Waltham, MA, USA). Four hundred ng total DNA from the extractions were applied for real-time PCR to amplify a portion of the FeHV-1 gB gene, using the procedure described in Section 2.5. Each sample analysis was performed in triplicate.

### 2.9. Virus Neutralizing Antibody Testing

Sera from all cats were analyzed for the presence of FeHV-1 virus neutralizing (VN) antibodies as previously described [25]. Briefly, sera were heat-inactivated at 56 °C for 30 min, followed by preparation of a two-fold serial dilution series. One hundred TCID_50_ of FeHV-1 C27 was mixed with the samples. Following a 1-h incubation at 37 °C, 2 × 10^4^ CRFK cells were added to each well and the plates were incubated for 3 days at 37 °C, 5% CO_2_. Each sample was tested in duplicate.

### 2.10. Cytokine mRNA Gene Expression Analysis

Total RNA was extracted from whole blood samples preserved in PAXgene^®^ blood RNA tubes (IVD), according to the manufacturer’s protocol (PreAnalytix GmbH, Hombrechtikon, Switzerland). Total RNA concentration was measured with a Nanodrop spectrophotometer (Thermo Fisher Scientific, Waltham, MA, USA) and reverse transcription was performed using 100 ng of sample RNA combined with 20 μL of a High-Capacity cDNA Reverse Transcription Kit and RNase Inhibitor (Applied Biosystems). Sequences of primers and probes for real- time PCR for glyceraldehyde 3-phosphate dehydrogenase (GAPDH), interferon alpha (IFNα), interferon beta (IFNβ), interferon gamma (IFNγ), tumor necrosis factor alpha (TNFα), interleukin 1 beta (IL-1β), interleukin 10 (IL-10), interleukin 12 subunit p40 (IL12p40), transforming growth factor beta (TGFβ), and chemokine C-C motif ligand 5 (CCL5, also known as RANTES) are listed in Table 4 and have been previously described [26,27,28,29,30]. The cDNA was diluted 1:20 for real-time PCR analysis and a 20 μL PCR cocktail was prepared, using 10 μL of TaqMan Fast Universal PCR Master Mix no AmpErase^TM^ UNG (Applied Biosystems), 0.8 μL of each primer (400 nM), 0.2 μL of probe (200 nM), 3.2 μL of sterile water, and 5 μL of cDNA. Glyceraldehyde 3-phosphate dehydrogenase (GAPDH) was used as an endogenous control for each gene of interest [18,31]. Real time PCR was performed using the 7500 Fast Real-Time PCR System with 7500 Software v2.0.6 (Applied Biosystems) The cycling conditions were as follows: 95 °C for 30 s, followed by 40 cycles consisting of 95 °C for 3 s and 60 °C for 30 s. Each sample was analyzed in triplicate and average CT values were corrected by subtracting the average CT of the housekeeping gene in the same sample.

### 2.11. Gross Pathology and Histological Examination

A gross pathological evaluation of animals was performed at the time of necropsy by a pathologist. Samples of nasal mucosa, tonsil, trachea, lung, retropharyngeal lymph node, heart, brain, liver, kidney, spleen, adrenal glands, testes, and ocular specimens, were collected, fixed in 10% buffered formalin, paraffin embedded, sectioned and routinely stained with hematoxylin and eosin. All sections were examined by a pathologist to evaluate pathological changes.

### 2.12. Statistical Analysis

For statistical analysis, all data was tested for normality using (GraphPad Prism Software v6). Where possible, not normally distributed data was transformed to achieve normal distribution. Data that was not normally distributed was analyzed by non-parametric tests. Total clinical scores for each period after vaccination and the period after challenge are shown as median scores for each period. The data was not normally distributed and was analyzed by Kruskal–Wallis multiple comparison test followed by Dunn’s post hoc multiple comparison test. For viral genome quantification assays in nasal swab samples, the data was normally distributed, and a repeated-measures one-way ANOVA was used to compare post-infection data overall between groups (GraphPad Prism Software v6). In addition, groups were compared on each day post-vaccination and post-infection using a two-way ANOVA Tukey’s post hoc multiple comparison test. The virus isolation data was not normally distributed and differences between groups were analyzed with a Friedman test. Viral genome data in trigeminal ganglionic samples was not normally distributed, and a Kruskal-Wallis test followed by Kolmogorov–Smirnov test was performed to compare (Cq) values from different groups (GraphPad Prism Software v6). For VN antibody titers, the data was log transformed to normalize the data and a two-way ANOVA Tukey’s post hoc multiple comparison test was used for analysis (GraphPad Prism Software v6). The cytokine/chemokine mRNA gene expression data was normally distributed. For statistical analysis, a Tukey’s multiple comparison test was used to compare differences on each day with pre-vaccination cytokine mRNA expression and differences in cytokine mRNA expression between groups (GraphPad Prism Software v6). A *p* value of <0.05 was regarded as significant difference for all data.

## 3. Results

### 3.1. Vaccination with the gE-TK- or PK- Mutants Was Safe and Significantly Reduced Clinical Scores Post Challenge Infection

Total median clinical scores per group are shown pre-vaccination (preV), post-V1 (day 0–21), post-V2 (day 21–42), post-V3 (day 42–63), and post-challenge (post-CH) periods (Figure 1A). All groups had a clinical score of zero during the pre-vaccination period. Similarly, following each vaccination with the PK- or gE-TK- mutants, median scores were 0. In the commercial vaccine group, the median score was 0.5 following the primary vaccination in that group. Overall, clinical scores were not significantly increased in any group as a result of vaccination. During the post-CH period, the control group had a median total score of 33.5 (cumulative score of 143), and the commercial vaccine group had a median total score of 14 (cumulative score of 67) (Figure 1A). When looking at the clinical scores on individual days following challenge infection (Figure 1B), a bi-phasic nature of the clinical score was observed in both controls and commercially vaccinated cats. In the control group, the first peak was noted starting on day 3 post-CH. The second peak was seen starting on day 7 post-CH. Similarly, in the commercial vaccine group, the primary peak started on day 3 post-CH and was not significantly different from that in the control group on the same day. Later in the course of infection (starting on day 7 post-CH), median daily total scores were significantly lower in the commercial vaccine group compared to the control group (Figure 1B); however, overall clinical scores were not significantly different in the control group compared to the commercial vaccine group (Figure 1A). In contrast, the median total score post-CH was 1 in the gE-TK- group (cumulative score of 7) and the median total score was 0 (cumulative score of 4) in the PK- group (Figure 1). Overall, daily clinical scores of the gE-TK- and PK- groups post-CH were significantly lower than those of the controls (*p* < 0.0001 (gE-TK-); *p* < 0.0001 (PK-)) and the commercial vaccine group (*p* < 0.0109 (gE-TK-); *p* < 0.0032 (PK-)), but daily clinical scores between gE-TK- and PK- groups were not significantly different from each other (Figure 1B).

### 3.2. Vaccination with the gE-TK- or PK- Mutants Prevented Body Weight Loss

Following infection, 3 of 4 cats in the control group and 2 of 4 cats in the commercial vaccine group exhibited weight loss and the remaining cats in each group stopped gaining weight (Table 5). In contrast, cats vaccinated with the gE-TK- or PK- vaccines maintained or gained weight.

### 3.3. Low Level Vaccine Virus DNA Was Detected in Nasal Secretions after Intranasal Administration of the PK- or gE-TK- Mutants, but No Infectious Virus Was Isolated

After intranasal vaccination with the PK- mutant, viral DNA was detected in the nasal secretions for 10 days with Ct values between 39 and 22 (Figure 2A). Shedding of viral DNA ceased by day 62 prior to challenge infection. In contrast, significant shedding of viral DNA was not detected in the gE-TK- group after intranasal vaccination and viral DNA was only detected at Ct levels between 36 and 39 in 3 of 6 cats on individual days. As expected, viral DNA was not recovered from any of the cats vaccinated with the commercial vaccine or the control cats at any time prior to challenge infection, indicating no cross contamination or intranasal exposure to vaccine virus occurred in these groups. Further, no infectious virus was detected by VI from nasal secretions of any cat at any time prior challenge infection. This indicates that any detection of low-level viral DNA prior to challenge infection was likely do to the presence of non-infectious virus-antibody complexes in the PK- and gE-TK- vaccinates, as has been described previously [21].

### 3.4. Vaccination with the gE-TK- or PK- Mutants Reduced Viral Nasal Shedding Compared to Controls and Commercially Vaccinated Cats Following Challenge Infection

Cats in the control group shed viral DNA in nasal secretions for up to 3 weeks following infection and peak levels were detected on day 4 post infection (day 67 of the study) (Figure 2A). Cats in the commercial vaccine group shed significantly decreased overall levels of viral DNA compared to the controls (*p* = 0.02); however, the levels on individual days post-CH were only significantly different on individual days 4 (*p* = 0.024), 8 (*p* = 0.0309), and 10 (*p* = 0.0196). In contrast, levels were significantly higher in control cats overall than levels in the gE-TK- vaccinated cats (*p* < 0.001), as well as on individual days 2 (*p* = 0.0001), 4 (*p* < 0.0001), 6 (*p* = 0.0082), 8 (*p* < 0.0001), and 10 (*p* < 0.0001), and 12 (*p* = 0.0163) post-CH. Similarly, levels in the PK- vaccination group were significantly lower than in control cats overall (*p* < 0.0001) and on individual days 2 (*p* = 0.0008), 4 (*p* < 0.0001), 6 (*p* < 0.0001), 8 (*p* < 0.0001) and 10 (*p* < 0.0001), and 12 (*p* = 0.0029) post-CH. Overall shedding post-CH was also significantly higher in the commercial vaccine group when compared to those in the gE-TK- group (*p* = 0.05) and PK- group (*p* = 0.005). Finally, shedding of viral DNA could still be observed in 3 of 4 control cats and 3 of 4 commercially vaccinated cats at the time point of necropsy. In contrast, only 1 of 6 cats in the gE-TK- or PK- vaccinated groups showed low level shedding of viral DNA at the time of necropsy. In addition to viral DNA quantification, VI on nasal swab samples was performed to test for infectious virus shedding (Figure 2B). Concurrent with the detection of viral DNA in nasal secretions, levels of VI overall were significantly different between groups. Specifically, more cats in the control group shed infectious virus following infection than any other group with 3 cats out of 4 shedding infectious virus on day 4 and 8 post-CH, and all four cats in the control group shedding infectious virus on Day 6 post-CH. In contrast, two cats in the commercial vaccine group shed infectious virus on day 6 post-CH and only one cat in the gE-TK- group shed infectious virus on day 6 post-CH. None of the cats in the PK- group shed infectious virus at any time point. As described for viral nasal shedding post vaccination, the discrepancy between the results of viral nasal shedding detected by real-time PCR (which detects viral DNA) and virus isolation (which detects infectious virus) is likely do to the neutralization of low-level replicating virus by VN antibodies, as described by Vögtlin et al. [21].

### 3.5. Differentiation between Vaccine Virus and Challenge Virus

For differentiation between vaccine virus and challenge virus, a conventional PCR based on the size differences due to gene deletions was carried out (Appendix A). Virulent FeHV-1, strain C-27, contained a full length of each gene (1867 base pairs (bp) for the gE gene, 1006 bp for the TK gene, and 1266 bp for the PK gene). The gE-TK- mutant contains a deficient gE gene of 268 bp, and a deficient TK gene of 543 bp. To differentiate between PK- vaccine virus and challenge virus, a conventional PCR amplifying the FeHV-1 PK gene was done on nasal swab samples collected from the PK- group, the commercial vaccine group, and the control group, on day 69 (day 6 post-CH) and on samples from the PK- group collected on day 48 (day 6 post IN vaccination) (Appendix A). All cats in the commercial vaccine group and the control group, in addition to 3 of 6 cats in the PK- vaccine group showed the full-size PK band on the gel post-CH (Appendix A). The remaining three cats in the PK- group did not show a band on day 6 post-CH. In contrast, the nasal swab samples from the PK- group collected on day 48 (day 6 post intranasal vaccination), showed a product of 379 bp in size, which indicates that the viral DNA shed at this point was the PK- vaccine virus.

To differentiate between gE-TK- vaccine virus and virulent challenge virus, conventional PCRs amplifying the FeHV-1 gE and TK genes were carried out on nasal swab samples from the gE-TK- group, commercial vaccine group, and control group, collected on day 69 (day 6 post-CH) and on day 48 (day 6 post IN vaccination on the 2 cats of the gE-TK- vaccine group that were positive by real-time PCR) (Appendix A). Samples from all cats in the gE-TK- group and all cats in the commercial vaccine group, and control group showed a PCR product of 1867 bp on day 6 post challenge, suggesting the virus present was the challenge virus. No PCR products were detected in any of the cats in the gE-TK- group following the intranasal vaccination presumably due to the absence or very low level of vaccine virus DNA in nasal secretions as a result of vaccination. The conventional PCR targeting the FeHV-1 TK gene showed a PCR product size of 1006 bp in 5 of 6 gE-TK- vaccinated cats and all cats in the commercial vaccine and control groups on day 6 post-CH, confirming the virus present in the samples was the challenge virus (Appendix A). Similar to the findings with the gE specific PCR, no TK-related PCR products were detected in any of the cats in the gE-TK- group following the intranasal vaccination, presumably due to no or very low-level vaccine virus DNA in nasal secretions as a result of vaccination.

### 3.6. Vaccination with the gE-TK- or PK- Mutants Significantly Reduced Viral DNA in the Trigeminal Ganglia (TG) Compared to Controls Following Challenge Infection

The results of viral quantification assays performed in trigeminal ganglionic samples are shown in Figure 3. The unvaccinated control group had a mean Ct value of 29.74, which was significantly lower than the value of 34.27 in the gE-TK- group (*p* = 0.05), or the value of 35.18 in the PK- group (*p* = 0.01) indicating higher levels of FeHV-1 gB DNA in the TG. There was no significant difference between controls and the commercial vaccine group (mean Ct value = 31.53). Further, Ct values in the commercial vaccine group were not significantly different from Ct values in the gE-TK- mutant group but were significantly lower than Ct-values in the PK- mutant group (*p* = 0.04). Finally, differences between gE-TK- and PK- groups were not significantly different and one cat was completely negative for FeHV-1 gB DNA, (Ct value = 40), in the gE-TK- group.

### 3.7. VN Titers Increased in All Vaccine Groups as a Result of Vaccination and the Increase in Post-Challenge VN Antibody Titer Was Highest in the Commercial Vaccine Group

Virus neutralization antibody titers following vaccination and challenge infection are shown in Figure 4. VN antibody titers of cats in the gE-TK- mutant group were significantly higher than in any other group on day 21, 3 weeks post-first-SC inoculation (*p* < 0.0001 compared to control group, *p* < 0.0001 compared to commercial vaccine group, *p* = 0.0114 compared to PK- group). In contrast, VN antibody titers in the PK- group on day 21 were not significantly different from the control or the commercial vaccine groups. On day 35, which is 14 days after the second SC inoculation with the gE-TK- or PK- mutant, or 14 days after first SC inoculation with the commercial vaccine, antibody titers in the gE-TK- group were not significantly different from those in the commercial vaccine group. In contrast, titers in the PK-vaccinated group were lower on day 35 than titers in the commercial vaccine group (*p* = 0.0192) and the gE-TK- group albeit not quite significant (*p* = 0.0755). Moreover, the titers in cats of the PK- group on day 35 were not significantly different from those of the cats in the control group (*p* = 0.1347). Just before the IN vaccination with the gE-TK- or PK- mutants or the second SC vaccination of the commercial vaccine group on day 42, antibody titers in all three vaccination groups were significantly higher than titers in the controls. However, titers in the gE-TK- vaccine group and in the commercial vaccine group were significantly higher than VN titers in the PK- vaccinated cats (*p* = 0.0196 and 0.0311). By day 56, titers in all vaccination groups were significantly higher than titers in controls but not significantly different from each other and this was also true on day 63 right before challenge infection.

On day 77, which is 14 days after challenge infection, titers in cats from the control group were not statistically different from titers of cats in the gE-TK- group (*p* = 0.5995) or cats in the PK- group (*p* = 0.7868). In contrast, the average titer of cats in the commercial vaccine group was significantly higher than those of the controls, gE-TK- group and PK- group (*p* = 0.0002, 0.0046, and 0.0016).

### 3.8. Cytokine mRNA Gene Expressions in Blood Following Vaccination and Challenge Infection

Whole blood mRNA gene expression for cytokines and chemokines were evaluated on day 0 before any treatment, two days after the second SC inoculation with the PK- or gE-TK- mutant in the PK- or gE-TK- vaccinated group (day 23), two days after the second SC inoculation with the commercial vaccine in the commercially vaccinated cats, or the IN inoculation with the PK- or gE-TK- in the PK- or gE-TK- vaccinated cats (day 44), and two days after challenge with virulent FeHV-1 in all groups (day 65), and results are shown in Figure 5 and Figure 6.

On day 23, which is two days after the second SC inoculation with the PK- or gE-TK- mutant, IFNα expression was increased in the PK- group (*p* = 0.0474). In addition, IFNβ mRNA expression was increased in both PK- and gE-TK- groups, compared to those of respective groups on day 0 (*p* = 0.0045 and 0.0137). In contrast, the expression of IL10 was significantly reduced in the gE-TK- group on day 23, (*p* = 0.0399), and there was a significant difference of IL10 expression between PK- and gE-TK- groups on day 23 (*p* = 0.0483). No significant difference was noted on day 23 for PK- and gE-TK- groups compared to levels on day 0, for IFNγ, IL1β, CCL5, TNFα, and TGFβ mRNA expression.

On day 44, which is two days after the IN inoculation with the PK- or gE-TK- mutants, or two days after the second SC inoculation with the commercial vaccine, there was a significant increase in IFNβ expression in the PK- group compared to that of the same group on day 0 (*p* = 0.045) and an increase in IFNβ expression in the gE-TK- group and the commercial vaccine group albeit not statistically significant. In addition, IL1β expression in the gE-TK- group was significantly lower than in the commercial vaccine group (*p* = 0.0283). Finally, IL10 was significantly reduced in all vaccine groups although only the levels in the gE-TK- vaccinated cats and in the commercially vaccinated cats were statistically significant, compared to levels of the respective group on day 0 (*p* = 0.0033 and <0.0001).

On day 65, which is two days after challenging with virulent FeHV-1, IFNα expression was increased in the commercial vaccine group and gE-TK- vaccinated group, compared to their pre-experimental levels on day 0 (*p* = 0.0287 and 0.0259, respectively). Coinciding, an increase in IFNβ expression was noted in the PK-, gE-TK-, and commercial vaccinated cats, compared to day 0 (*p* = 0.0228, 0.0014, and 0.0055). Moreover, the increases of IFNβ in the gE-TK- group on day 65 was significant when compared to its counterparts on day 23 (*p* = 0.0072) and day 44 (*p* = 0.0219). The type II interferon, IFNγ was also increased in the PK-, gE-TK-, and commercially vaccinated cats when compared to their counterparts on day 0 (*p* = 0.0025, 0.0041, and 0.0023), in addition to an increase in CCL5 expression in the gE-TK- vaccinated cats compared to the levels on day 0 (*p* = 0.0323). IL1β expression was significantly increased in all four groups when compared to day 0 (*p* = 0.0271 for PK- group, *p* < 0.0001 for gE-TK- group, *p* = 0.002 for commercial vaccine group, and *p* = 0.0009 for controls). However, the increases in IL1β expression were higher in the commercial vaccine group or the controls than in the PK- or gE-TK- group (*p* = 0.0084 and 0.0119, respectively compared to commercial vaccine cats) (*p* = 0.0014 and 0.0021, respectively compared to control). A significant increase was also noted for TNFα expression in the PK- group (*p* = 0.0018) and the gE-TK- group (*p* = 0.0004) and a similar trend was observed for TGFβ expression in the PK- group (*p* < 0.0001), the gE-TK- group (*p* < 0.0001), and the control group (*p* < 0.0001). Interestingly, TGFβ expression was significantly lower in the commercial vaccine group compared to the controls (*p* = 0.0137). TGFβ expression was also higher in the PK- vaccinated cats post-challenge when compared to post-vaccination levels on day 23 (*p* < 0.0001) or on day 44 (*p* = 0.001), and in the gE-TK- vaccinated cats when compared to day 23 (*p* < 0.0001). Finally, a significant increase in IL10 expression was observed in the PK- group on day 65 compared to that on day 0 (*p* = 0.042). On the contrary, a significant decrease in IL10 expression was seen in the commercial vaccine group (*p* = 0.0002) and the controls (*p* = 0.0123) compared to their levels on day 0.

### 3.9. Vaccination with gE-TK- and PK- Reduced Lesions in the Lungs and Nasal Turbinates Compared to Lesions Observed in Cats of the Commercial Vaccine Group and the Controls

Necropsies and gross examinations were conducted 3 weeks post infection, when all cats had clinically recovered. Gross lesions related to FeHV-1 infection were not detected in any of the cats. Nasal turbinate, tonsil, trachea, lung, retropharyngeal lymph node, heart, brain, liver, kidney, spleen, adrenal glands, testes, and ocular specimens from all cats were examined microscopically. Lesions in all cats were limited to the lungs and nasal turbinates, and microscopic images are shown in Figure 7. All cats in the control group (*n* = 4) had congested lungs with a mild interstitial pneumonia and moderate numbers of foamy macrophages in the alveoli, mild alveolar edema and few interstitial neutrophils and lymphocytes. Two cats had moderate bronchus associated lymphoid tissue (BALT) hyperplasia (Figure 7A). All four cats had a moderate chronic lymphoplasmacytic rhinitis with formation of lymphoid follicles in two cats and ulceration of the mucosal epithelium with the necrotic surface being covered by degenerate neutrophils in one animal (Figure 7B). In the commercial vaccine group (*n* = 4), all four cats had few intra-alveolar macrophages and edema fluid in the lungs, two cats had BALT hyperplasia and one cat had a mild interstitial pneumonia with mild perivascular lymphoid infiltrates (Figure 7C). In addition, three cats out of four in the commercial vaccine group had a mild to moderate rhinitis with erosions and suppurative inflammation in one cat (Figure 7D). In contrast, histological changes in the gE-TK- and PK- groups were much milder, especially in the nasal turbinates. Lungs of all cats in the gE-TK- group (total *n* = 6) had some intra-alveolar edema fluid and few intra-alveolar macrophages (Figure 7E) and three cats had mild BALT hyperplasia. Only one cat had a mild rhinitis, while most cats had a normal nasal mucosa (Figure 7F). In the PK- group all cats (total *n* = 6) had similar changes as the cats in the gE-TK- group and three cats had mild BALT hyperplasia in the lung. Two cats had a mild focal interstitial pneumonia, but overall, the lungs were unremarkable (Figure 7G). Only one of the six PK- cats had a mild rhinitis while the remaining cats had an unremarkable nasal mucosa (Figure 7H). No microscopic lesions related to this study were noted in other organs or tissues.

## 4. Discussion

The current study shows safety and efficacy of a novel vaccination regimen that utilizes a gE-TK- FeHV-1 deletion mutant or a PK-FeHV-1 deletion mutant administered twice subcutaneously (SC) followed by IN mucosal boosting in SPF kittens. Our vaccination regimen was chosen based on the hypothesis that adding a mucosal boost to the standard two dose regimen of SC immunization would induce protective immunity not only against clinical signs but also against virus shedding after challenge. This hypothesis is based on data that shows that boosting via the intranasal route following priming via the SC route has a synergistic effect on reduction of viral shedding post-challenge by stimulating robust mucosal and systemic immunity [20,32]. Reagan et al. [20] further showed that kittens vaccinated concurrently via the SC and IN routes were better protected against clinical signs and virus shedding of virulent virus, administered at 1 week post vaccination. Priming via the SC route, followed by an IN boost has also been recommended for cats older than 2 months without maternal antibody protection [33].

In our study, we show that clinical protection was significantly improved using our vaccination regimen with either the gE-TK- mutant or the PK- mutant when compared to unvaccinated controls or cats vaccinated with a commercial MLV vaccine. Specifically, the cumulative clinical scores after challenge were extremely low in both the PK- and gE-TK- groups (score of 7 and 4, respectively), with clinical signs limited to fever for 1 day in 3/6 cats (gE-TK- group), sneezing for 1–3 days in 2/6 cats in the gE-TK- group and 1/6 cats in the PK- group, and mild nasal congestion for 1 day in 1 cat in the PK- group. In contrast, cats that were not vaccinated or cats immunized with a commercial vaccine prior to challenge exhibited fever, sneezing, coughing, nasal discharge, nasal congestion, blepharospasms, and ocular discharge for up to 17 days post-challenge (total scores 143 in controls and 67 in commercial vaccine group). Furthermore, 3/4 cats and 2/4 cats exhibited weight loss and the remaining cats stopped gaining weight following challenge infection in unvaccinated or commercial vaccine cats, while all cats vaccinated with the gE-TK- or PK- mutants maintained or continued gaining weight. The histological findings supported the clinical data after challenge infection and showed that all cats in the control group and 2/4 cats in the commercial vaccine group had moderate to severe interstitial pneumonia and 3/4 cats showed rhinitis, while the cats in the gE-TK- and PK- groups had milder or no lesions in lungs and nasal turbinates.

In addition to the reduction of clinical disease, challenge virus DNA levels in nasal secretions were significantly lower in the groups immunized with the PK- and gE-TK- mutants than in those that were either not vaccinated or vaccinated with a commercial vaccine. Furthermore, virus isolations indicated that none of cats shed virulent challenge virus in the group immunized with the PK- mutant, while only one cat showed infectious viral shedding on day 6 post-challenge in the gE-TK- group. These data parallel those of a vaccination-challenge study that included a pseudorabies virus (PRV) PK- deletion mutant. This mutant completely protected pigs from respiratory shedding of infectious virus [11]. However, experimental immunization with gE-TK- double deletion mutants during in vivo testing for (PRV) and bovine herpesvirus-1 (BHV-1) did not completely prevent infectious virus shedding post challenge [11,12].

We previously reported that PK- and gE-TK- mutants of FeHV-1 were attenuated and showed significantly reduced viral replication in feline respiratory epithelial cells [18], which was taken as an in vitro indicator for being a candidate for further in vivo testing. The data from the present study confirmed that the presence of gE-TK- viral DNA in nasal secretions after IN inoculation stayed extremely low (Ct value no less than 35) in 4/6 cats or was not detectable (2/6 cats) during the whole observation period following vaccination. In contrast, viral DNA was detected in all six of the cats in the PK- mutant group post-intranasal vaccination with an average peak (Ct = 30) on day 6 post intranasal vaccination. However, in perspective, infectious virus could not be detected by virus isolation from any of the collected nasal secretions samples in either of these groups during the observation period following vaccination. This indicates that, despite two previous SC immunizations, the PK- mutant could still replicate at a very low level, which was detectable by PCR but not by VI because it was either below the limit of detection of VI or complexed with antibody. Additionally, while there was low level replication detected by PCR, this replication was not associated with clinical signs except for one cat that showed mild sneezing on three occasions following the intranasal vaccination.

Our safety and efficacy data using the gE-TK- or PK- mutant as a mucosal booster following SC priming indicate an improvement in safety over previous studies that reported some residual virulence following mucosal administration only with a FeHV-1 gE-gI deletion mutant at higher dose levels in cats. This higher dose level correlated however with the level of protection from virulent FeHV-1 challenge [14,25]. In another study, intranasal administration of a single TK deletion mutant of FeHV-1 in cats resulted in sneezing and, following challenge, significant virus shedding was detected in oral, nasal, and ocular secretions [15]. It is important to recognize that, while we did show that our vaccination regimen with the gE-TK- and PK- mutants was safe and efficacious, further studies are needed to directly evaluate the safety of the mutants when given intranasally without prior SC priming or compare mucosal boosting in previously vaccinated cats. Further, it is important to note that our experimental design does not allow us to differentiate between the effect of the vaccine itself and the effect of the changes in the vaccine regimen. However, current vaccines are not licensed for mucosal administration for safety reasons and the goal of this study was to simply develop an attenuated vaccine that could safely be administered as part of a mucosal vaccine regimen and show that this strategy is superior to what is currently being used in feline practice.

A further advantage of using deletion mutants is the ability to differentiate FeHV-1 vaccine strains from virulent field strains, which can be a challenge with conventional vaccines where the genomes of vaccine virus and virulent virus are nearly identical [34]. In the present study, the shedding of virulent FeHV-1 post-challenge was easily differentiated from that of the deletion mutants via PCR testing targeting the PK, gE, or TK genes. The results obtained showed that the viral shedding post-challenge in PK- and gE-TK- groups consisted completely of virulent FeHV-1 challenge virus. This is also critical for eradication programs that rely on differentiating animals that are immune due to vaccination from infected animals.

Fever was never observed in the group immunized with the PK- mutant, either after vaccination or after the challenge infection. In contrast, occasional fever responses lasting 1 day were noted in 3/6 cats immunized with the gE-TK- mutant following the primary or secondary SC immunization, but no clinical signs were noted after intranasal vaccination in this group. Fever can be an indicator of an ongoing systemic immune response and is often associated with the development of VN antibody titers following immunization [20,35] and the lack of fever in the PK- mutant immunized group correlated with the lower slope of the VN antibody titer curve following SC immunization in this group. Further, there was a sharp post-challenge spike of the VN antibody titer in the group immunized with the commercial vaccine, which was not observed in the groups immunized with either the PK- or gE-TK- deletion mutants or the control groups. This spike in the commercial vaccine group may be a reflection of significant post-challenge virulent virus shedding in this group, not observed in the gE-TK- or PK- vaccine groups.

In addition to VN antibodies, expression of type I interferons is a major antiviral immune response in the defense against viral infection. Activation of IFN-regulatory factor 3 (IRF3) and NF-κB initiate expression of IFNα, while IRF7 combined with NF-κB initiate the expression of IFNβ. Both the PK and TK genes of alphaherpesviruses have shown to modulate the activation of IFNα and IFNβ in blood leucocytes by reducing TLR3 expression or by counteracting subsequent transcription factor IRF3 [36,37,38,39]. In the present study, we confirmed an induction of type 1 interferons in the blood following vaccination with the PK- or gE-TK- mutants, especially IFNβ, which was increased two days after the second SC inoculation with the deletion mutants. Such an increase was not found in the group immunized with a commercial vaccine at any time point. On the other hand, the gene expression of IFNγ, which is a type II interferon, was not altered in any of the vaccinated groups until the challenge occurred.

Consistent with an increase in interferons, expression of IL10 was significantly decreased in the gE-TK- vaccine group after the second SC inoculation and IN inoculation. This decrease was not shown to the same degree in the PK- group. It has been shown in a HSV-1 study that alphaherpesviruses manipulate T cell activation by inhibiting the activation of the T cell signaling complex and the associated NF-κB activation, while the p38 kinase and c-Jun N-terminal kinase (JNK) is activated by the virus. Consequently, the virus-infected T cells could selectively synthesize IL10 to favor immune suppression by the virus [40]. Moreover, it is believed that IL10, along with IL4, IL5, and IL6, is a Th2 cytokine that favors antibody development and B cell activation, in contrast to Th1 cytokines such as IFNγ and TNFα, which induce the activation of macrophages and T cells [41]. IL10 is also known to inhibit activation of Th1 responses by suppressing the ability of antigen-presenting cells to secrete Th1-associated cytokines [42], which eliminate herpesvirus infection. We speculate that the decrease in IL10 by gE-TK- vaccination in our study might have contributed to the activation of cellular immunity and type 1 interferons by the mutant.

An apparent increase in cytokine expression including IFNs, TGFβ, and TNFα, was further observed in all groups after challenge infection. However, the level of IL1β post-challenge was lower in the groups immunized with the PK- or gE-TK- mutants than in the group immunized with a commercial vaccine or the control group. IL1β is an inflammatory cytokine and it is likely that the increased levels observed in the groups immunized with a commercial vaccine or the unvaccinated control group reflected the increased challenge virus replication and airway inflammation compared to the gE-TK- and PK- vaccination groups. Similarly, IL1β was one of the key cytokines correlating with the severity of HSV1-associated encephalitis and neural invasion in vivo [43]. The increased IL1β levels in the controls and commercial vaccinates in our study that coincide with increased viral replication and inflammation might also explain the observed increase in viral DNA in the TG. Similarly, efficient HSV-1 replication at the primary infection site is associated with high viral copy numbers in neuronal cell bodies (Thompson et al., 2000). This is further supported by our results showing increased viral DNA detection in trigeminal ganglia of controls and commercially vaccinated cats. This is also compatible with a previous study showing that the quantity of FeHV-1 DNA found in trigeminal ganglia at 30 days post-inoculation was significantly correlated with the total clinical scores [24]. However, the detection of FeHV-1 latency-associated transcripts in the trigeminal ganglia from our cats has not yet been performed and this data would provide further direct evidence of latency establishment [44]. Interesting are also reports linking decreased levels of latent herpesvirus DNA with a decreased propensity for viral reactivation [45]. In addition, there are reports showing that strategies that reduce acute viral shedding also reduce the establishment of latency and the frequency of reactivation [46,47].

Current commercial immunization for FeHV-1 consists of two consecutive subcutaneous (SC) inoculations [4]. A vaccination regimen which consists of inoculating the same antigen via different administration routes for priming and subsequent boosting is not a standard procedure. A dual-site immunization strategy has been deployed against a variety of viral infections, such as human immunodeficiency virus [48], poliovirus [49], and measles virus [50]. It was found to induce more effective protection by stimulating stronger humoral and cellular immunity [51,52]. Further, the addition of vaccine administration via the IN route induces local immune responses, such as IgM and IgA as well as local cytokine and chemokine responses at the mucosal level [53]. In mice, it has been reported that immunization regimens that include a mucosal route resulted in a significant production of mucosal IgA and a significant secretion of IL17, which is a pro-inflammatory cytokine critical for CD4^+^ T cell activation and Th1 polarization [54]. In our study, we specifically aimed at improving induction of mucosal immunity by incorporating IN administration with the PK- and gE-TK- into our vaccination regimen, but further investigations of IgA or cytokine/chemokine quantification in nasal secretions are necessary to verify this assumption. The present study suggests a benefit of an additional boost via a mucosal route over standard dual SC immunization, albeit a separate study using the selected mutants and comparing a SC only vaccination regimen with a combination of a SC/IN prime boost regimen would have to be performed to ultimately proof this hypothesis.

## 5. Conclusions

In conclusion, the present study consists of a comprehensive safety and efficacy evaluation of an immunization regimen consisting of two SC immunizations followed by a IN administration of two pilot vaccine candidates (PK- and gE-TK- deletion mutant of FeHV-1). The immunization regimen was compared to the currently recommended vaccination regimen for commercial vaccines, which consists of two SC immunizations. Parameters evaluated in this study included safety, induction of immune responses, and protection from challenge with a clinical FeHV-1 isolate. Our data clearly indicate the benefit of the selected vaccination regimen with attenuated FeHV-1 mutants in terms of enhancing protection against clinical disease, virus shedding and latency.

## Figures and Tables

**Figure 1 viruses-13-00163-f001:**
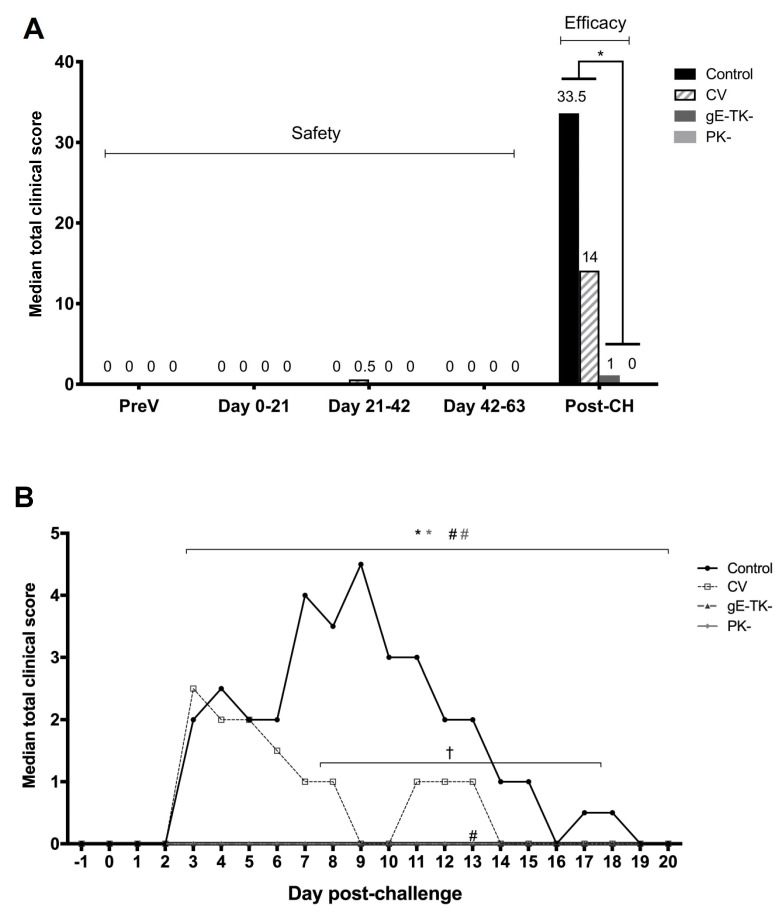
Clinical scores. (**A**) Median total clinical scores of controls (black bars), commercial vaccine group (CV, dashed bars), gE-TK- vaccinated group (dark grey bars), and PK- vaccinated group (light grey bars). Median total scores pre-vaccination (pre-V), from day 0 to 21 (post-V1 for gE-TK- and PK-), from day 21 to 42 (post-V2 for gE-TK- and PK- and post-V1 for CV), from day 42 to 63 (post-V3 for gE-TK- and PK- and post-V2 for CV), and post-challenge (post-CH) from day 64 to day 83. An Asterisk represents a significant difference (*p* < 0.05); (**B**) Median total daily scores of controls are shown from one day before challenge to day 20 post-challenge infection. Black * and grey *, respectively, indicate a significant difference (*p* < 0.05) between the controls and the gE-TK- vaccinates and the PK- vaccinates. Black # and grey #, respectively, indicate a significant difference (*p* < 0.05) between the commercial vaccine group and the gE-TK- vaccinates and the PK- vaccinates. Black † indicates a significant difference (*p* < 0.05) between the commercial vaccine group and the controls.

**Figure 2 viruses-13-00163-f002:**
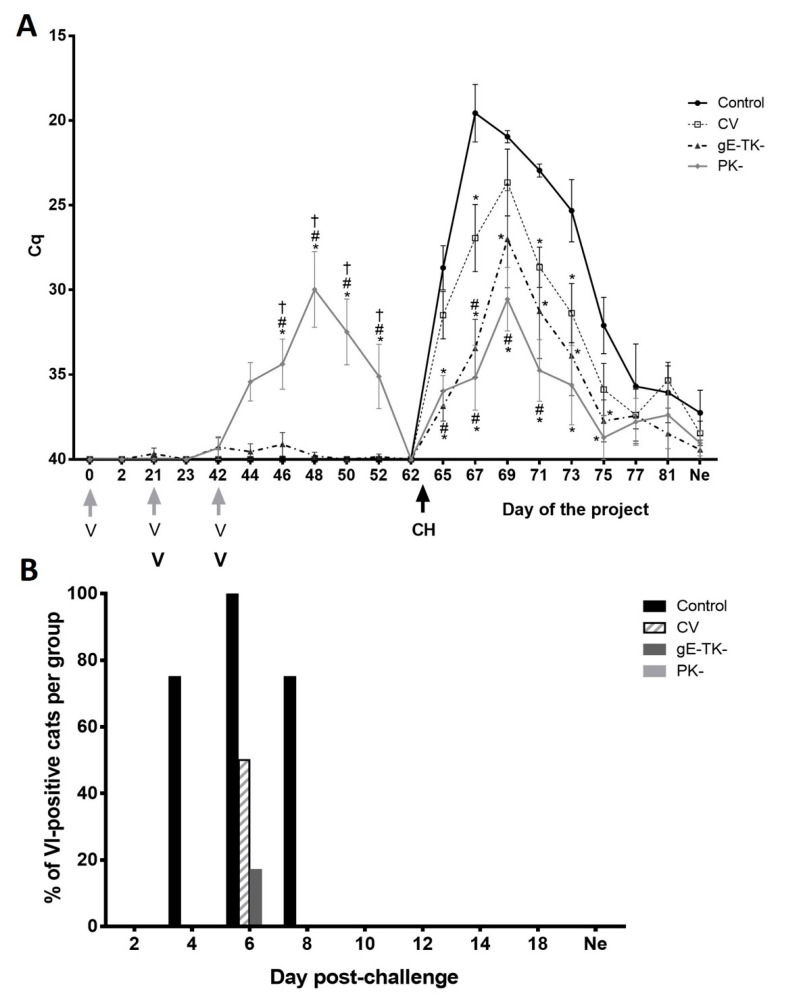
Viral nasal shedding: (**A**) Presence of FeHV-1 gB DNA quantitated by real-time PCR. Average viral shedding ±SEM (standard error of the mean) of controls (black circle and solid lines), commercial vaccine group (CV, open squares and dotted lines), gE-TK- group (triangles with dashed lines), and PK- group (diamonds with grey lines). * represents a significant difference (*p* < 0.05) from the control group. # represents a significant difference (*p* < 0.05) from the commercial vaccine group. † represents a significant difference (*p* < 0.05) between PK- and gE-TK- groups. Vaccinations in the gE-TK- and PK- groups on days 0, 21, and 42 are indicated by grey arrows and “V”. Vaccinations in the CV group on days 21, and 42 are indicated by a black bolded “V”. The challenge infection on day 63 is indicated by a black arrow and “CH”; (**B**) Virus isolation following challenge infection. The percentages of cats that were positive by virus isolation are indicated by black bars (controls), dashed bars (commercial vaccine group, CV), dark grey bars (gE-TK- group), and light grey bars (PK- group) daily post-CH.

**Figure 3 viruses-13-00163-f003:**
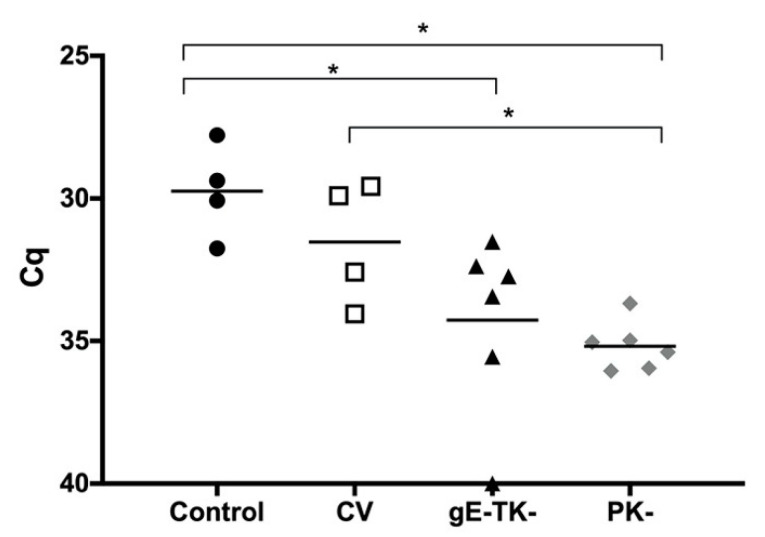
Viral gB DNA presence in trigeminal ganglia. The line is the mean Ct value for each group; Symbols represent individual animals. Black circles are controls, white squares are conventional vaccinates, black triangles are gE-TK- vaccinates and gray diamonds are PK- vaccinates. * Represents a significant difference (*p* < 0.05).

**Figure 4 viruses-13-00163-f004:**
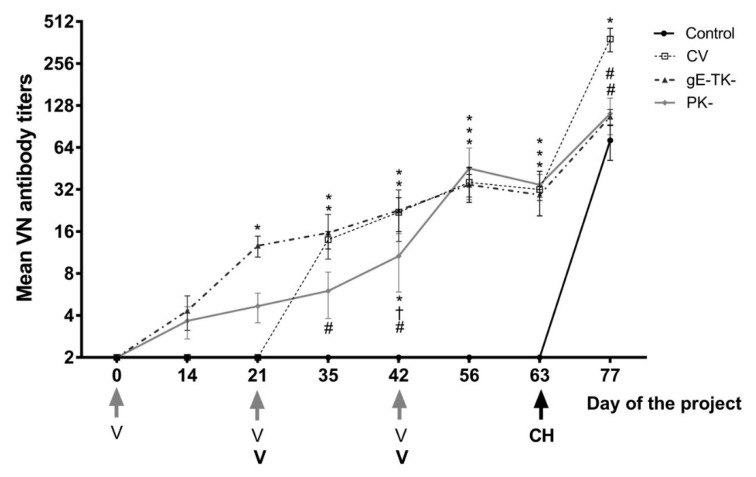
Average virus neutralization antibody titers ±SEM (standard error of the mean) of controls (black circle and solid lines), commercial vaccine group (CV, open squares and dotted lines), gE-TK- group (triangles with dashed lines), and PK- group (diamonds with grey lines). * represents a significant difference (*p* < 0.05) from the control group. # represents a significant difference from the commercial vaccine group. † represents a significant difference (*p* < 0.05) between the PK- and gE-TK- groups. Vaccinations of the gE-TK- and PK- group on day 0, 21, and 42 are highlighted by grey arrows (V). Vaccinations of the CV group are indicated by a black “V”. The challenge infection on day 63 is indicated by a black arrow and CH.

**Figure 5 viruses-13-00163-f005:**
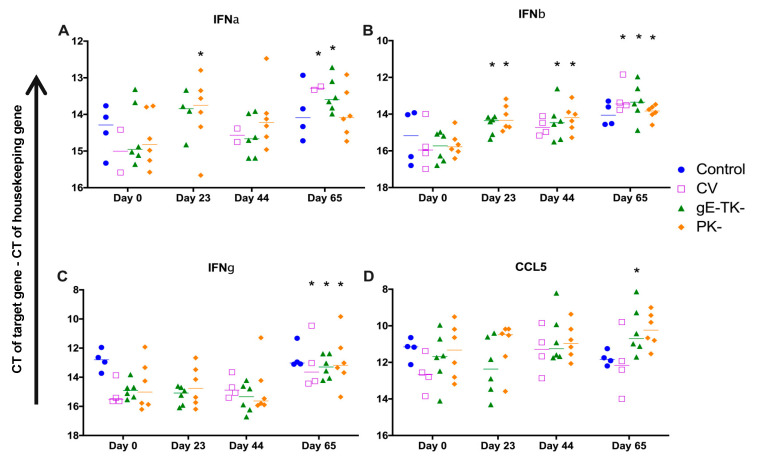
Interferon and CCL5 mRNA expression in whole blood samples: IFNα (**A**); IFNβ (**B**); IFNγ (**C**); CCL5 (**D**). The line is the median of CT values for each group corrected by a housekeeping gene (GAPDH) and individual symbols represent corrected CT values for individual cats. * represents a significant difference (*p* < 0.05) from the same group from day 0. CV: Commercial vaccine group. Time points of vaccinations were on days 0, 21 and 42 for the gE-TK- and PK- vaccinates and on days 21 and 42 for the commercial vaccine group.

**Figure 6 viruses-13-00163-f006:**
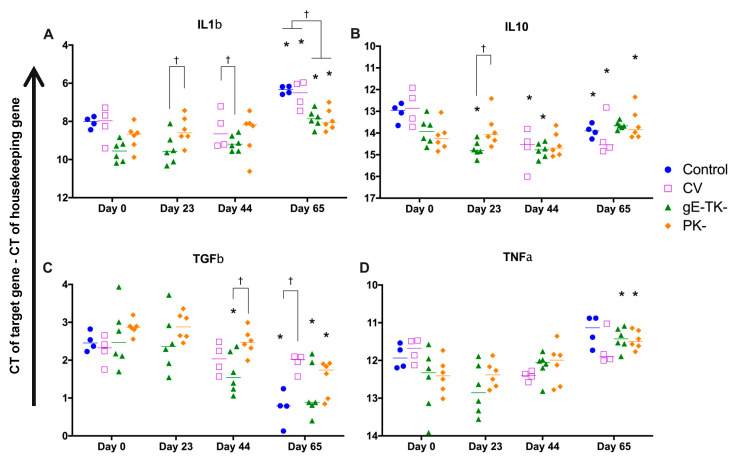
Cytokine mRNA expression in whole blood samples: IL1b (**A**); TNFα (**B**); TGFβ (**C**); IL10 (**D**). The line is the median of CT values for each group corrected by a housekeeping gene (GAPDH) and individual symbols represent corrected CT values for individual cats; * Represents a significant difference (*p* < 0.05) from the same group from day 0. † represents a significant difference (*p* < 0.05) between groups on the same sampling day. CV: Commercial vaccine group. Time points of vaccinations were on days 0, 21 and 42 for the gE-TK- and PK- vaccinates and on days 21 and 42 for the commercial vaccine group.

**Figure 7 viruses-13-00163-f007:**
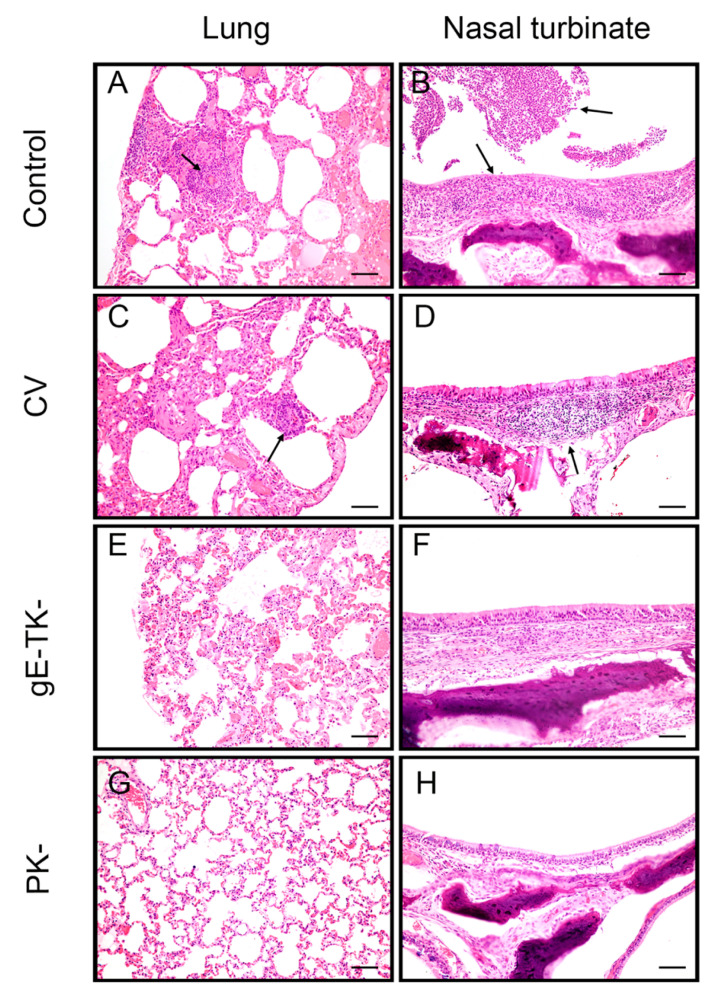
Histological changes in lung and nasal turbinates at time of necropsy. Representative images of the lungs from controls (**A**); commercial vaccine group (CV) (**C**); gE-TK- group (**E**); PK- group (**G**). Representative images of nasal turbinates from controls (**B**); commercial vaccine group (**D**); gE-TK- group (**F**); PK- group (**H**). Arrows indicate the infiltration and aggregation of inflammatory cells, composed of lymphocytes and neutrophils. The size of the scale bar is 150 μm.

**Table 1 viruses-13-00163-t001:** Timeline and experimental design of the study. The cats in the gE-TK- and PK- groups were subject to subcutaneous inoculations with gE-TK- and PK- mutants, respectively, on days 0 (gray V1) and 21 (gray V2), followed by an intranasal inoculation with the respective mutant on day 42 (gray V3). The commercial vaccine group was vaccinated twice subcutaneously with the commercial vaccine on day 21 (black V1) and 42 (black V2). No inoculation was conducted in the control group. All four groups were challenge infected with virulent FeHV-1 (CH) intranasally on day 63. The solid line represents daily physical exams. Dotted lines represent nasal swab collection which was done every other day. “X” indicates specific days a collection was performed. Ne: necropsy day.

	Day of the Project
−7	0	2	4	14	21	23	25	35	42	44	52	56	62	63	65	77	81	Ne
**Treatment**		V1				V2/V1				V3/V2					CH				
**Clinical score**		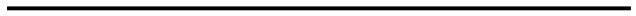
**Nasal shedding**	X	X	X	X	X	X	X	X	X	X 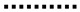 X	X	X		X 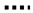 X	X	X
**VN test**		X			X	X			X	X			X		X		X		
**Blood cytokine**		X					X				X					X			
**Histology**																			X

**Table 2 viruses-13-00163-t002:** Clinical score criteria.

Category	Signs	Score
Conjunctivitis	Normal	0
	Mild hyperemia	1
	Moderate to severe hyperemia	2
	Moderate to severe hyperemia, with chemosis	3
Blepharospasm	Normal	0
	<25% of eye closed	1
	25–50% of eye closed	2
	50–75% of eye closed	3
	75–100% of eye closed	4
Ocular discharge	Normal	0
	Minor, serous	1
	Minor to moderate, mucoid	2
	Marked mucopurulent	3
Sneezing	Normal	0
	Sneezing observed or evident with nasal Discharge with blood on walls of cage	1
Nasal discharge	Normal	0
	Minor, serous, occasional with blood	1
	Minor to moderate, mucoid or bloody	2
	Marked mucopurulent	3
Nasal congestion	Normal	0
	Mild	1
	Moderate	2
	Severe, with open mouth breathing	3
Coughing	Normal	0
	Coughing noted	1
Fever	Body temperature lower or equal to 39.2 °C	0
	Body temperature higher than 39.2 °C	1

**Table 3 viruses-13-00163-t003:** Primers for differentiating FeHV-1 WT and the mutants.

Viral Gene	Forward Primer (5′-3′)	Reverse Primer (5′-3′)	Expected Product Size
**PK**	AGGCACTCAGTGGGCCAAAGT	AGGCTGTCTTACACATGAGGCA	WT: 1266 bp PK-: 379 bp
**gE**	GGTCATGTGTAATGTTGACG	ATACAATATACGCGTTTGACG	WT: 1867 bp gE-TK-: 268 bp
**TK**	ACGATGGCGAGTGGAACCATCC	TACGGTGCATATCACATGCGGCTA	WT: 1006 bp gE-TK-: 543 bp

**Table 4 viruses-13-00163-t004:** Primers and probes for cytokine gene expression assays.

	Forward Primer (5′-3′)	Reverse Primer (5′-3′)	Probe (5′-3′)
**GAPDH**	GCCGTGGAATTTGCCGT	GCCATCAATGACCCCTTCAT	CTCAACTACATGGTCTACATGTTCCAGTATGATTCCA
**IL1β**	AATGACCTGTTCTTTGAGGCTGAT	CCAGAAAACTGTGGCTCAGGTT	CGAAAAGATGAAGGGCAGCCTCCAA
**IL10**	ACTTTAAGGGTTACCTGGGTTG	CGTGCTGTTTGATGTCTGG	TTGGAGGAGGTGATGCCCCA
**IL12p40**	TGGCTTCAGTTGCAGGTTCTT	TGGACGCTATTCACAAGCTCA	CGGTTTGATGATGTCCCTGATGAAGAAGCT
**IFNα**	CTTGACGCTCCTGGGACAAA	ACTGGTCTCCACCAGAACACG	TCCCTGCCATCTCCTGTCAGAAGG
**IFNβ**	TGGAATGAGACCACTGTTGAGAA	GGATCGTTTCCAGGTGTTCCT	CTCCTTGCGACACTCCACTGGCAG
**IFNγ**	TGCAAGTAATCCAGATGTAGCAG	GTTTTATCACTCTCCTCTTTCCAG	CAAAATGTCTACGAAAAGCGACCCACC
**TNFα**	CACATGGCCTGCTGCAACTAATC	AGCTTCGGGGTTTGCTACTAC	TCTCGAACTCCGAGTGACAAGCCA
**TGFβ**	GGAATGGCTGTCCTTTGATG	TGCAGTGTGTTATCTTTGCTGTC	TTTCGCCTCAGTGCCCACTG
**CCL5**	CTACACCAGCAGCAGTGTTCC	ACACACCTGGCGCTTCCTC	TGCCCAGCAGTCGTCTTTGTCACCC

**Table 5 viruses-13-00163-t005:** Body weight changes. Weight loss compared to the previous observation is bolded and shaded in dark grey.

	Cat	Control				CV				gE-TK-						PK-					
Day		67	68	69	70	71	72	73	74	75	76	77	78	79	80	81	82	83	84	85	86
**−4**		2.0	2.2	2.1	2.4	3.0	2.4	1.9	2.1	2.2	2.4	2.2	2.1	2.2	2.1	2.1	2.3	2.2	2.0	2.8	2.4
**0**		2.1	2.3	2.1	2.4	2.9	2.4	2.0	2.1	2.3	2.4	2.2	2.0	2.0	2.0	2.2	2.5	2.3	2.2	2.7	2.5
**6**		2.4	2.6	2.3	2.8	3.2	2.6	2.2	2.5	2.5	2.7	2.4	2.3	2.4	2.4	2.5	2.9	2.4	2.2	2.8	2.8
**14**		2.7	2.9	2.7	3.1	3.0	3.0	2.1	2.5	2.7	3.0	2.7	2.5	2.5	2.6	2.9	3.0	2.8	2.4	3.3	2.9
**21**		3.0	3.2	2.9	3.3	3.6	3.2	2.6	2.8	2.9	3.3	2.8	2.6	2.7	2.8	2.9	3.2	2.9	2.5	3.4	3.1
**35**		3.4	3.6	3.2	3.2	3.9	3.6	3.0	3.4	3.3	3.8	3.2	2.8	2.9	3.1	3.2	3.5	3.4	2.8	3.9	3.5
**42**		3.6	3.8	3.5	4.1	4.2	3.8	3.3	3.7	3.5	3.9	3.3	2.9	3.1	3.3	3.5	3.8	3.6	2.9	4.1	3.7
**44**		3.6	3.8	3.5	4.1	4.2	3.8	3.3	3.7	3.5	3.9	3.3	2.9	3.1	3.3	3.5	3.8	3.6	2.9	4.1	3.7
**56**		4.0	4.4	3.7	4.5	4.5	4.2	3.4	3.9	3.9	4.3	3.6	3.2	3.2	3.6	3.8	4.1	3.9	3.2	4.5	4.1
**63**		4.1	4.5	3.9	4.8	4.7	4.3	3.8	4.2	4.0	4.6	3.7	3.4	3.4	3.7	4.0	4.2	4.2	3.4	4.7	4.3
**72**		**3.9**	**4.2**	**3.5**	4.8	4.8	**3.9**	**3.7**	4.2	4.2	4.7	3.7	3.5	3.5	3.7	4.1	4.4	4.3	3.5	4.9	4.4

## Data Availability

The data are contained within the article or Appendix A.

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
