# Peer review of "Safety and Efficacy of Felid Herpesvirus-1 Deletion Mutants in Cats"

_viruses, 2021, doi:10.3390/v13020163_

Round 1

Reviewer 1 Report

This manuscript reports on a new vaccination strategy for FHV-1 in cats. It compares a commercial vaccine with two pilot vaccine candidates (PK- and gE-TK- deletion mutant of FHV-1) and provides convincing data for their superior efficacy and reduced subsequent viral shedding etc.

There are a number of issues in the text that need to be addressed. These are listed below.

Abstract

Line 21: It would be good to change the wording slightly, as it is not possible that “unvaccinated controls” “were vaccinated”.

Line 22: Replace “disease” by “signs” or similar.

Line 24: The abbreviations “gE-TK-“ and “PK-” should be introduced to be understood.

Line 25: Replace “post mortem pathology” by a more appropriate term, such as “pathological changes”.

Line 27/28: “modulation of … and IL10” needs rewording. The authors should at least add the word “transcription” after “IL10”.

Introduction

Lines 35 and 44: Add “infection” after “FHV-1”.

Line 46: Replace “symptoms” by “signs”.

Line 48: The reviewer is not aware that avirulent FHV-1 latency exists and would therefore suggest to delete “virulent”, unless the reviewer is wrong.

Line 52: The statement in the sentence requires a reference.

Line 53; Delete the first “of”.

Line 80: Change to “Experimental”.

Materials and methods

Line 114: Add information on the country where the manufacturer is based.

Line 120: Replace “begin” by a full word, such as “onset” or “beginning”.

Line 121: Replace “the time of necropsy” by “during necropsy” or similar, to obtain a correct sentence.

Line 152: Add “of” after “because”.

Line 154: Add “associated disease” or similar after “FHV-1”.

Line 157: The sentence is incomplete (after “60 minutes”).

Line 193: Is there a special reason for mentioning the catalogue number of the filters but no such number for any other consumable?

Line 196. The authors mention a “characteristic cytopathic effect”; they should add a reference for this.

Line 232: Delete “In addition”.

Lin2 236: The authors should specify that the pathologist undertook a histological examination to determine any pathological changes (or similar wording).

Results

Line 268: Please reword to make clear that the clinical scores did not differ significantly. The way it is worded at present, the authors say that the clinical scores were not statistically significant…

Line 275: Add “the” in front of “control”.

Line 280-283: Please check the sentence, it looks like s.th. has happened during the editing process.

Line 283: Replace “between” by “in”.

Line 310: Add “to” after “prior”.

Line 316: The wording “on overall” does not seem to be correct English.

Lines 318, 320: Replace “control” by “control animals” or “control cats”.

Line 327: If the authors want to use the abbreviation “VI” also in the results section and further (like in line 310), they should be consistent and also do this in line 327 instead of reintroducing the abbreviation.

Line 329: Replace “was” by “were”.

Lines 331, 332: Is “in the control group” needed here? Looks like an unnecessary repetition.

Line 333: Add “the” after “in”.

Line 340: Delete “in size”.

Lines 342-350: There is s.th. wrong, part of the text is repeated three times. This needs to be corrected.

Line 367: Delete “on samples”.

Line 380: Change the wording; “…reduced presence…” is equivocal.

Line 384: Add “the” twice after “in”.

Line 385: Was the abbreviation “TG” introduced? And if so, is it necessary?

Line 399: Reword; “those of cats” does not make sense.

Line 407: Reword “titer of PK- group” and “that of control group” and replace “were” by “was”.

Line 411: Add “group” after “vaccine”.

Line 434, 436, 443, 447, 450, 452, 454, 456, 457, 459, 461, 463, 470, 471: The authors speak frequently about up- and downregulation. This is an interpretation of higher and lower transcription levels and suggests an active process that has not been demonstrated. Therefore, the authors should avoid these terms when describing the findings re. the relative cytokine transcription levels.

Line 484/485: The heading is not well worded. Change to reflect what was observed.

Line 491: Replace “were shown” by “show”.

Line 491, 497, 503: When the authors write “congested lungs” this implies passive hyperaemia, a finding that should be graded to some extent (mild, moderate, severe) and would be expected to occur due to euthanasia. It has its histological equivalent in the more general term “hyperemia”. The pathologist should decide whether this is an agonal change or a pathological finding in the animals in the present study and interpret/present it accordingly.

Line 496: The authors describe ulceration. Did they check for the presence of FHV-1 antigen and i.n. inclusion bodies in epithelial cells? This could in principle be a virus-associated lesion.

Line 500: See previous comment. It could also be valid for the “erosions” described here.

Line 501: Replace “findings” by “changes”.

Line 506: Replace “lung tissues” by “the lung”.

Discussion

Line 545: The wording “unremarkable lesions” should be changed, as it is meaningless – either there were lesions or there were none, with their presence the degree might differ.

Line 596: Delete the first “in the”.

Line 605: Replace “upregulated” by “higher”.

Line 611: Replace “downregulated” by “lower”.

Line 624: Add “apparent” after “An” (see previous comments).

Line 633: The authors should specify what they refer to when they speak about “increase in neurotropism in the present study”.

Tables:

Table 1: The horizontal lines appear to be in the wrong position (too low?).

Table 2: The term “nasal congestion” should be explained with regard to the clinical/pathological process this implies. Congestion itself means “passive hyperemia” and is probably not referred to by the authors in the current context (see also line 535). For the body temperature the authors now refer to Fahrenheit, but otherwise to °C. They should be consistent, and °C is preferred.

Figures:

Figure 1A: Change the label on the Y axis to “Median total clinical score” (similar to Fig 1B) so that the reader can see what is shown.

Figure 2: The first sentence of the legend needs to be reworded. “Viral shedding” does not take place “in nasal swab samples”…

In line 356, add an explanation for SEM (like has been done for Figure 4).

Figure 7: The histological images appear to be of relatively poor quality. The resolution should be improved and it might be preferable to use images of a higher magnification.

Supplemental figures:

It looks like there is no reference to Supplemental figure 1. Should this not be provided?

Author Response

Reviewer 1:

This manuscript reports on a new vaccination strategy for FHV-1 in cats. It compares a commercial vaccine with two pilot vaccine candidates (PK- and gE-TK- deletion mutant of FHV-1) and provides convincing data for their superior efficacy and reduced subsequent viral shedding etc.

There are a number of issues in the text that need to be addressed. These are listed below.

Abstract

Line 21: It would be good to change the wording slightly, as it is not possible that “unvaccinated controls” “were vaccinated”.

Corrected in the manuscript and highlighted in red

Line 22: Replace “disease” by “signs” or similar.

Changed to “signs” and highlighted in red in the text

Line 24: The abbreviations “gE-TK-“ and “PK-” should be introduced to be understood.

This was added in line 18

Line 25: Replace “post mortem pathology” by a more appropriate term, such as “pathological changes”.

Corrected in the manuscript and highlighted in red

Line 27/28: “modulation of … and IL10” needs rewording. The authors should at least add the word “transcription” after “IL10”.

 This was changed to “In addition, after challenge infection we observed a reduction of IL-1β expression, and modulation of TNFα, TGFβ and IL10 expression”

Introduction

Lines 35 and 44: Add “infection” after “FHV-1”.

Corrected in the manuscript and highlighted in red

Line 46: Replace “symptoms” by “signs”.

Corrected in the manuscript and highlighted in red

Line 48: The reviewer is not aware that avirulent FHV-1 latency exists and would therefore suggest to delete “virulent”, unless the reviewer is wrong.

Corrected in the manuscript and highlighted in red

Line 52: The statement in the sentence requires a reference.

A reference was added.

Line 53; Delete the first “of”.

I could not find any extra “of” in lines 52-55

Line 80: Change to “Experimental”.

Corrected in the manuscript and highlighted in red

Materials and methods

Line 114: Add information on the country where the manufacturer is based.

Corrected in the manuscript and highlighted in red

Line 120: Replace “begin” by a full word, such as “onset” or “beginning”.

Corrected in the manuscript and highlighted in red

Line 121: Replace “the time of necropsy” by “during necropsy” or similar, to obtain a correct sentence.

Corrected in the manuscript and highlighted in red

Line 152: Add “of” after “because”.

Corrected in the manuscript and highlighted in red

Line 154: Add “associated disease” or similar after “FHV-1”.

Corrected in the manuscript and highlighted in red

Line 157: The sentence is incomplete (after “60 minutes”).

Corrected in the manuscript and highlighted in red

Line 193: Is there a special reason for mentioning the catalogue number of the filters but no such number for any other consumable?

No, this was an error. We corrected this in the manuscript and highlighted it in red

Line 196. The authors mention a “characteristic cytopathic effect”; they should add a reference for this.

A reference was added

Line 232: Delete “In addition”.

Corrected in the manuscript and highlighted in red

Lin2 236: The authors should specify that the pathologist undertook a histological examination to determine any pathological changes (or similar wording).

 Corrected in the manuscript and highlighted in red

Results

Line 268: Please reword to make clear that the clinical scores did not differ significantly. The way it is worded at present, the authors say that the clinical scores were not statistically significant…

The intent was to point out that there was no significant clinical disease due to vaccination. The sentence has been reworded to reflect this.

Line 275: Add “the” in front of “control”.

Corrected in the manuscript and highlighted in red

Line 280-283: Please check the sentence, it looks like s.th. has happened during the editing process.

Corrected in the manuscript and highlighted in red

Line 283: Replace “between” by “in”.

Corrected in the manuscript and highlighted in red

Line 310: Add “to” after “prior”.

Corrected in the manuscript and highlighted in red

Line 316: The wording “on overall” does not seem to be correct English.

Corrected in the manuscript and highlighted in red

Lines 318, 320: Replace “control” by “control animals” or “control cats”.

Corrected in the manuscript and highlighted in red

Line 327: If the authors want to use the abbreviation “VI” also in the results section and further (like in line 310), they should be consistent and also do this in line 327 instead of reintroducing the abbreviation.

Corrected in the manuscript and highlighted in red

Line 329: Replace “was” by “were”.

Corrected in the manuscript and highlighted in red

Lines 331, 332: Is “in the control group” needed here? Looks like an unnecessary repetition.

“in the control group” was deleted

Line 333: Add “the” after “in”.

Corrected in the manuscript and highlighted in red

Line 340: Delete “in size”.

This line was taken out and changed as part of addressing the comment on providing a link to supplemental figure 1. All changes in this section 3.5 are highlighted in red

Lines 342-350: There is s.th. wrong, part of the text is repeated three times. This needs to be corrected.

Corrected in the manuscript and highlighted in red

Line 367: Delete “on samples”.

Corrected in the manuscript and highlighted in red

Line 380: Change the wording; “…reduced presence…” is equivocal.

Corrected in the manuscript and highlighted in red

Line 384: Add “the” twice after “in”.

Corrected in the manuscript and highlighted in red

Line 385: Was the abbreviation “TG” introduced? And if so, is it necessary?

Corrected in the manuscript and highlighted in red

Line 399: Reword; “those of cats” does not make sense.

Corrected in the manuscript and highlighted in red

Line 407: Reword “titer of PK- group” and “that of control group” and replace “were” by “was”.

Corrected in the manuscript and highlighted in red

Line 411: Add “group” after “vaccine”.

Corrected in the manuscript and highlighted in red

Line 434, 436, 443, 447, 450, 452, 454, 456, 457, 459, 461, 463, 470, 471: The authors speak frequently about up- and downregulation. This is an interpretation of higher and lower transcription levels and suggests an active process that has not been demonstrated. Therefore, the authors should avoid these terms when describing the findings re. the relative cytokine transcription levels.

Corrected in the manuscript and highlighted in red

Line 484/485: The heading is not well worded. Change to reflect what was observed.

The heading was re-worded and is highlighted in red in the manuscript

Line 491: Replace “were shown” by “show”.

The sentence was re-worded and is highlighted in red in the manuscript

Line 491, 497, 503: When the authors write “congested lungs” this implies passive hyperaemia, a finding that should be graded to some extent (mild, moderate, severe) and would be expected to occur due to euthanasia. It has its histological equivalent in the more general term “hyperemia”. The pathologist should decide whether this is an agonal change or a pathological finding in the animals in the present study and interpret/present it accordingly.

The reviewer is correct to point out the differences between hyperemia and congestion. As all animals were euthanized, congestion of the lungs would be expected due to the failing heart and slight variations in severity could easily be attributed to differences in the response of individual cats to the euthanasia procedure. Histologically both congestion and hyperemia are characterized by distention of vessels, especially the capillary bed with red blood cells. While hyperemia in the alveoli in response to inflammation tends to be more patchy and more prominent than typical congestion, in cases of mild to moderate accumulation of blood without the evidence of vascular wall damage, e.g. leaking of fluid, fibrin etc. differentiating between congestion and hyperemia is very difficult, if not impossible.  This is the reason why in toxicologic pathology neither congestion nor hyperemia are used as histologic scoring criteria (see terminology used by National Toxicology Program/NIEHS: https://ntp.niehs.nih.gov/nnl/respiratory/lung/index.htm). As the primary changes caused by the FeHV-1 infection are microscopically reflected through differences in the degree of inflammation and edema, we deleted the few references regarding congestion as the observed differences of vascular dilation were not the primary difference between groups. If the reviewer would like to see this information included, we could use the descriptive term of vascular bed dilation rather than interpreting the change as hyperemia versus congestion.

Line 496: The authors describe ulceration. Did they check for the presence of FHV-1 antigen and i.n. inclusion bodies in epithelial cells? This could in principle be a virus-associated lesion.

The reviewer is correct that these lesions are most likely caused by FeHV-1. We did not see inclusions and in our experience finding herpesviral inclusions in affected surface epithelia is uncommon as the infected cells will quickly slough off into the lumina. As for IHC, the purpose of pathology was to compare morphologic alterations. IHC for FeHV-1 to associate typical viral lesions with viral antigen was negative this late in the course of infection (>21 days post CH). This is not surprising as VI also confirmed infectious virus was only present until day 8 post CH. For quantitative purposes, PCR provided more objective means to score viral load and shedding over the whole time period post CH.

Line 500: See previous comment. It could also be valid for the “erosions” described here.

See above.

Line 501: Replace “findings” by “changes”.

Corrected in the manuscript and highlighted in red

Line 506: Replace “lung tissues” by “the lung”.

Corrected in the manuscript and highlighted in red

Discussion

Line 545: The wording “unremarkable lesions” should be changed, as it is meaningless – either there were lesions or there were none, with their presence the degree might differ.

Corrected in the manuscript and highlighted in red

Line 596: Delete the first “in the”.

Corrected in the manuscript and highlighted in red

Line 605: Replace “upregulated” by “higher”.

Corrected in the manuscript and highlighted in red

Line 611: Replace “downregulated” by “lower”.

Corrected in the manuscript and highlighted in red

We have further changed all the up-and downregulated to increased/decreased respectively in the whole section

Line 624: Add “apparent” after “An” (see previous comments).

Corrected in the manuscript and highlighted in red

Line 633: The authors should specify what they refer to when they speak about “increase in neurotropism in the present study”.

This statement refers to the higher levels of viral DNA in the TG in controls and CV. The wording has been changed and a clarification has been added in the manuscript

Tables:

Table 1: The horizontal lines appear to be in the wrong position (too low?).

The lines have been moved

Table 2: The term “nasal congestion” should be explained with regard to the clinical/pathological process this implies. Congestion itself means “passive hyperemia” and is probably not referred to by the authors in the current context (see also line 535). For the body temperature the authors now refer to Fahrenheit, but otherwise to °C. They should be consistent, and °C is preferred.

This has been changed in the table

Figures:

Figure 1A: Change the label on the Y axis to “Median total clinical score” (similar to Fig 1B) so that the reader can see what is shown.

The label has been changed

Figure 2: The first sentence of the legend needs to be reworded. “Viral shedding” does not take place “in nasal swab samples”…

This has been changed to “Viral nasal shedding” and the in nasal swab samples has been taken out.

In line 356, add an explanation for SEM (like has been done for Figure 4).

Corrected in the manuscript and highlighted in red

Figure 7: The histological images appear to be of relatively poor quality. The resolution should be improved and it might be preferable to use images of a higher magnification.

The images that are pasted into word are of low resolution. However, I have submitted the original, which is of good quality, as a separate file.

Supplemental figures:

It looks like there is no reference to Supplemental figure 1. Should this not be provided?

A reference to supplemental figure 1 was added at the beginning of section 3.5. and is highlighted in red.

Reviewer 2 Report

Dear Editor,

Feline Herpesvirus 1 (FeHV-1) is an important respiratory pathogen of cats. The World Small Animal Veterinary Association (WSAVA) has considered that anti-FeHV-1 vaccine is a core vaccine for cats, meaning that every cat should be vaccinated against this virus. Unfortunately, in contrast with modified live virus vaccines used as core vaccines in dogs or in contrasts with the anti feline parvovirus vaccine, vaccines against FeHV-1 do not offer long lasting immunity, do not prevent infection, virus shedding and latency establishment. New vaccines are therefore needed.

In their submitted manuscript, the authors pursue their work on two candidate FeHV-1 vaccines that they are developing. The first one is a strain deleted for gE and TK, the second one is deleted for PK. In this study, the authors compared the capacity of these vaccines to induce a protective immune response in cats with commercial vaccines. They also looked at virus shedding after challenge, titres of neutralizing antibodies, cytokine mRNA gene expression and post-mortem histology and detection of latency establishment. They nicely showed that these viral strains apparently induce better immune response and significantly reduced clinical scores, post-mortem lesions, virus shedding and latency establishment.

The paper is well written and the experiments are nicely done, especially considering the difficulty to perform experiments on cats. However, this is difficult to compare results generated with different experimental protocols. Indeed, the cats immunized with the commercial vaccine (PUREVAX) received subcutaneous injection of this MLV vaccine on days 21 and 42 according to the manufacturer recommendations. In contrast, cats that were vaccinated with the two deleted strains received subcutaneous injections on days 0 and 21 and were then boosted intranasally on day 42. The timing of vaccination, the number of doses and the route of administration are therefore different. It appears therefore impossible to distinguish between an improvement due to the vaccine strains and an improvement due to the vaccination procedure. What would have been the results if the commercial vaccine would have been administered following the same scheme as the two tested strains?

Especially, the authors compare in figure 4 the neutralizing antibody levels at day 21 between cats vaccinated with these two strains and those immunized with the commercial vaccine! Unless misunderstanding, the cats immunized with CV received their first administration on that day!

Finally, the authors mention in the introduction that numerous reports of virulence of commercial vaccines when administered intranasally. This is likely but this would have been especially interesting to show that the two strains tested in this study have advantages on these commercial vaccines when administered intranasally. Without any experimental data produced at the same time this remains speculative.

Major comments:

  • In their experimental design, the authors compare different viruses administered through the same regimen (gE-TK- and PK-) to a commercial vaccine administered through another scheme. This is interesting, but how is it possible to differentiate the effect of the vaccine from the effect of the regimen?
  • The results obtained in Figure 3 suggest that latency establishment is lower in cats vaccinated with the two tested strains. This could be interesting, however, what does it mean. Are subtle levels in latency loads really important? The main message here is that while the level of latency is lower, latency is not abolished.
  • In figure 4, the authors compare the neutralizing antibody titres in cats vaccinated at different dates!
  • Similarly, figures 5 and 6 are very difficult to analyse based on those differences in treatment. Moreover, what is the relevance of these differences of values expressed as fold changes. It therefore requires a very difficult interpretation to understand what it means. This would be easier to provide raw data.
  • The differences in the route of administration could also largely influence the mucosal lesions.

Minor comments:

  • The authors use the abbreviation FHV-1 for the virus. However, based on ICTV recommendations, should it not be FeHV-1?
  • In the introduction, the authors describe the advantages of several deletion mutants of alphaherpesviruses as vaccine strains but they do not talk about their use as marker vaccines associated with DIVA tests. This of particular importance for PRV and BoHV-1 and should therefore be briefly discussed.
  • Lines 89 and following: the authors say that based upon characterization of the mutants… but they never describe what these features are. This should be described.
  • Line 98: why males?
  • Lines 304-310: the authors should better describe the results obtained in control group and in the commercial vaccine group.
  • Lines are repeated between lines 342 and 348. This chapter is therefore ot really understandable as written.
  • How to explain the apparent discrepancy between results of panels A and B of Figure 2.

Author Response

Reviewer 2:

Dear Editor,

Feline Herpesvirus 1 (FeHV-1) is an important respiratory pathogen of cats. The World Small Animal Veterinary Association (WSAVA) has considered that anti-FeHV-1 vaccine is a core vaccine for cats, meaning that every cat should be vaccinated against this virus. Unfortunately, in contrast with modified live virus vaccines used as core vaccines in dogs or in contrasts with the anti feline parvovirus vaccine, vaccines against FeHV-1 do not offer long lasting immunity, do not prevent infection, virus shedding and latency establishment. New vaccines are therefore needed.

In their submitted manuscript, the authors pursue their work on two candidate FeHV-1 vaccines that they are developing. The first one is a strain deleted for gE and TK, the second one is deleted for PK. In this study, the authors compared the capacity of these vaccines to induce a protective immune response in cats with commercial vaccines. They also looked at virus shedding after challenge, titres of neutralizing antibodies, cytokine mRNA gene expression and post-mortem histology and detection of latency establishment. They nicely showed that these viral strains apparently induce better immune response and significantly reduced clinical scores, post-mortem lesions, virus shedding and latency establishment.

The paper is well written and the experiments are nicely done, especially considering the difficulty to perform experiments on cats. However, this is difficult to compare results generated with different experimental protocols. Indeed, the cats immunized with the commercial vaccine (PUREVAX) received subcutaneous injection of this MLV vaccine on days 21 and 42 according to the manufacturer recommendations. In contrast, cats that were vaccinated with the two deleted strains received subcutaneous injections on days 0 and 21 and were then boosted intranasally on day 42. The timing of vaccination, the number of doses and the route of administration are therefore different. It appears therefore impossible to distinguish between an improvement due to the vaccine strains and an improvement due to the vaccination procedure. What would have been the results if the commercial vaccine would have been administered following the same scheme as the two tested strains?

We agree with the reviewer and have responded to the point of the experimental design under “Major comments” below. As for the time points in the figures we did line up the vaccinations, so that immunity resulting from each regime could at the very least directly be compared following the last vaccination and immediately prior to challenge infection between the groups.

Especially, the authors compare in figure 4 the neutralizing antibody levels at day 21 between cats vaccinated with these two strains and those immunized with the commercial vaccine! Unless misunderstanding, the cats immunized with CV received their first administration on that day!

It is correct that a comparison on day 21 makes only sense for the 2 vaccine candidates. As indicated in the response to the previous comment, for comparing immune responses between experimental groups, the only time points relevant are time points after day 42 (which is the time point of the last vaccine for each group). However, we felt that it was still of interest to report immune responses earlier, as it shows that there was no accidental exposure or cross contamination and cats responded with the responses one would expect after vaccination. In addition, there is some new information on how cats responded to our candidate vaccines after each vaccination and how the two candidate vaccines compared to each other. In addition, the comparative responses immediately prior to and after challenge infection are of interest and relevant. To clarify, we have added and additional “V” in a different color representing the vaccinations in the CV group and separating them from the gE-TK-  and PK- vaccine groups and added that information into the figure legend (see also below under major comments).

Finally, the authors mention in the introduction that numerous reports of virulence of commercial vaccines when administered intranasally. This is likely but this would have been especially interesting to show that the two strains tested in this study have advantages on these commercial vaccines when administered intranasally. Without any experimental data produced at the same time this remains speculative.

The reports that are mentioned in the introduction are already referenced in the manuscript. We agree that it would have been very interesting to include a CV vaccine group that was vaccinated with the same vaccine regimen used for the deletion mutants to have a direct comparison.  This was unfortunately not possible due to ethical and financial restraints (see also below under major comments) and thus some of our conclusions are speculative and we tried to reflect this in the language chosen throughout the manuscript.

Major comments:

  • In their experimental design, the authors compare different viruses administered through the same regimen (gE-TK- and PK-) to a commercial vaccine administered through another scheme. This is interesting, but how is it possible to differentiate the effect of the vaccine from the effect of the regimen?

We agree with the reviewer that the current study does not allow to differentiate  between the effect of the vaccine from the effect of the regimen. However, our intent was in the first instance to develop a regimen that would be a significant improvement over what is currently available on the market and is used in cats in practice. As we believe that stimulation of mucosal immunity is critical, we wanted to develop a vaccine that we could administer mucosally, which cannot be done with commercial vaccines for safety reasons. While it would have been ideal to have more experimental groups to differentiate the effect of the vaccine from the effect of the regimen, this was not possible due to ethical and financial reasons. As current vaccines are not licensed for intranasal administration, it is also unlikely that we would have gotten IACUC approval for using the commercial vaccine in this way. Thus, the goal of the current study was test whether we could use the selected FeHV-1 deletion mutants to develop a safe and efficacious subcutaneous prime- mucosal boost vaccination regimen that would be superior to currently commercially available FeHV-1 vaccination regimens in terms of reducing clinical signs, limiting virus shedding and latency, associated with inducing neutralizing antibodies and cytokine responses in specific-pathogen-free cats challenged with virulent FeHV-1. We believe that we have succeeded in this endeavor and we hope to leverage this data in the future to answer some of the questions raised here and further differentiate the effect of the vaccine from the effect of the regimen within the ethical limits of such studies.

We have included more information and changed the wording to reflect the purpose of our study more accurately and justify our experimental design in lines: 19-20, 68-69, 102-106, 573-574, 622-627, 704, 705, 712-719 and highlighted the sections in the manuscript in red font and green highlighter.

  • The results obtained in Figure 3 suggest that latency establishment is lower in cats vaccinated with the two tested strains. This could be interesting, however, what does it mean. Are subtle levels in latency loads really important? The main message here is that while the level of latency is lower, latency is not abolished.

While it is true that latency is not abolished, there are some reports that lower levels of latency are correlated with lower levels of viral nasal shedding and that for HSV-1 reduced levels of latent viral DNA correlates with a lower propensity for re-activation, so reducing levels of viral DNA in the TG could be relevant. We have added this information in the manuscript in line 689-692.

  • In figure 4, the authors compare the neutralizing antibody titres in cats vaccinated at different dates!

See as above:  A comparison on day 21 makes only sense for the 2 vaccine candidates. As far as comparing immune responses between experimental groups, the only time points relevant are any time points after day 42 (the time point of the last vaccine for each group). However, we felt that it was still of interest to report immune responses earlier, as it shows that there was no accidental exposure or cross contamination and cats responded with responses one would expect after vaccination. In addition, there is some new information on how cats responded to our candidate vaccines after each vaccination and how the two candidate vaccines compared to each other. Also, the comparative responses after challenge infection are of interest and relevant. To clarify, we have added and additional “V” in a different color representing the vaccinations in the CV group and separating them from the gE-TK-  and PK- vaccine groups in table 1 and Figures 2A and 4  and added that information into the figure legend in table 1 and Figures 1, 2A, 4, 5 and 6.

  • Similarly, figures 5 and 6 are very difficult to analyse based on those differences in treatment. Moreover, what is the relevance of these differences of values expressed as fold changes. It therefore requires a very difficult interpretation to understand what it means. This would be easier to provide raw data.

We have changed the figures and presented the data as raw CT’s corrected with a housekeeping gene. We also added the time point of vaccinations for each group to the legends.

  • The differences in the route of administration could also largely influence the mucosal lesions.

It is true that the reduction in mucosal lesions in the PK- and gE-TK- group is likely do to the mucosal vaccination and induction of mucosal immunity, which is unlikely to occur to the same extend with SC only vaccination of currently marketed vaccines. This is precisely why we believe that mucosal boosting will improve currently iused vaccination regimens.

Minor comments:

  • The authors use the abbreviation FHV-1 for the virus. However, based on ICTV recommendations, should it not be FeHV-1?

This has been changed throughout the manuscript

  • In the introduction, the authors describe the advantages of several deletion mutants of alphaherpesviruses as vaccine strains but they do not talk about their use as marker vaccines associated with DIVA tests. This of particular importance for PRV and BoHV-1 and should therefore be briefly discussed.

An additional paragraph has been added to the introduction in lines 69-73 and in the discussion in line 634-635.

  • Lines 89 and following: the authors say that based upon characterization of the mutants… but they never describe what these features are. This should be described.

This information was added to the introduction in lines 98 to 102.

  • Line 98: why males?

We typically select all males or all females because of difficulties that are encountered when housing intact male and female cats together and the increased stress levels involved for the cats as a consequence.

  • Lines 304-310: the authors should better describe the results obtained in control group and in the commercial vaccine group.

An explanation as to why no shedding of virus was detected or expected has been added to paragraph 3.3.

  • Lines are repeated between lines 342 and 348. This chapter is therefore ot really understandable as written.

This has been fixed in the text.

  • How to explain the apparent discrepancy between results of panels A and B of Figure 2

Detection of low level (high CT) viral nasal shedding by PCR in the absence of detection of infectious virus is not uncommon and is likely do to neutralization of low level replicating virus by VN antibodies as well as the presence of non-infectious viral DNA. This phenomenon has been previously described by Vögtlin et al. J Clin Microbiol. 2002 Feb; 40(2): 519–523, and we have added this information to paragraph 3.3 and 3.4.

Round 2

Reviewer 2 Report

Dear Editor,

Thank you for sending back the revised manuscript.

The authors addressed all my comments in a quite convincing way. However, they did not provide more comparisons with the actual commercial vaccines. They have to be very careful with their comparisons. I still think that it would have been more interesting to compare all vaccines with the same regimens. At the moment, this is very speculative and we cannot really be sure that either the commercial vaccines would induce so much side effects when administered intranasally (this is just based on other publications that are not so clear) or that the alternative strains do not have any side effect (very difficult to assess with so small numbers). Therefore, to my opinion, this study shows that these vaccines strains and this procedure are good alternatives to the current commercial vaccines, not that they are better.